# Plant Extracts as Possible Agents for Sequela of Cancer Therapies and Cachexia

**DOI:** 10.3390/antiox9090836

**Published:** 2020-09-07

**Authors:** Jinjoo Lee, Myung In Jeong, Hyo-Rim Kim, Hyejin Park, Won-Kyoung Moon, Bonglee Kim

**Affiliations:** 1College of Korean Medicine, Kyung Hee University, Hoegi-dong Dongdaemun-gu, Seoul 05253, Korea; leejinjoo1202@khu.ac.kr (J.L.); auddls07@khu.ac.kr (M.I.J.); hyorim5102@khu.ac.kr (H.-R.K.); hyejinpark46@khu.ac.kr (H.P.); wonkyung95@khu.ac.kr (W.-K.M.); 2Department of Pathology, College of Korean Medicine, Kyung Hee University, Hoegi-dong Dongdaemun-gu, Seoul 05253, Korea; 3Korean Medicine-Based Drug Repositioning Cancer Research Center, College of Korean Medicine, Kyung Hee University, Hoegi-dong Dongdaemun-gu, Seoul 05253, Korea

**Keywords:** cancer sequela, cachexia, plant extracts, antioxidants, reactive oxygen species, oxidative stress, inflammation

## Abstract

Cancer is a leading cause of the death worldwide. Since the National Cancer Act in 1971, various cancer treatments were developed including chemotherapy, surgery, radiation therapy and so forth. However, sequela of such cancer therapies and cachexia are problem to the patients. The primary mechanism of cancer sequela and cachexia is closely related to reactive oxygen species (ROS) and inflammation. As antioxidant properties of numerous plant extracts have been widely reported, plant-derived drugs may have efficacy on managing the sequela and cachexia. In this study, recent seventy-four studies regarding plant extracts showing ability to manage the sequela and cachexia were reviewed. Some plant-derived antioxidants inhibited cancer proliferation and inflammation after surgery and others prevented chemotherapy-induced normal cell apoptosis. Also, there are plant extracts that suppressed radiation-induced oxidative stress and cell damage by elevation of glutathione (GSH), superoxide dismutase (SOD), catalase (CAT), glutathione peroxidase (GPx) and regulation of B-cell lymphoma 2 (BcL-2) and Bcl-2-associated X protein (Bax). Cachexia was also alleviated by inhibition of tumor necrosis factor-α (TNF-α), interleukin-1β (IL-1β), interleukin-6 (IL-6) and monocyte chemoattractant protein-1 (MCP-1) by plant extracts. This review focuses on the potential of plant extracts as great therapeutic agents by controlling oxidative stress and inflammation.

## 1. Cancer Therapies and Cachexia

Cancer is a major public health problem worldwide and causes a high mortality rate of 53% [1]. Excluding non-melanoma skin cancer, 18.1 million cases of new cancer diagnosis and 9.6 million deaths were reported and the leading causes of cancer-related deaths in both sexes were lung cancer (18.4%), colorectal cancer (9.2%), stomach cancer (8.2%) and liver cancer (8.2%) [1]. To reduce the cancer morbidity, there have been various approaches to treat cancer. It includes surgery [2,3], chemotherapy [4], radiotherapy [5,6], immunotherapy [7], hormone therapy [8] and so forth.

### 1.1. Limitation of the Current Cancer Therapies

Some sequela can limit the benefits of these treatments. Surgery is followed by the risk of pain, infection and micro metastasis. Some surgeries also induce deterioration of the function of the organ under surgery. For example, about 75% of sixty-nine patients were reported that usual interstitial pneumonias were diagnosed after 60 lobectomies and 9 segmentectomies compared to the results of preoperative high-resolution computed tomography [9]. Chemotherapy and radiotherapy can damage normal cells and cause secondary cancer. They can also cause fatigue, anorexia, edema, fever for general disorders. There are gastrointestinal, blood, cardiac, nervous system, skin and respiratory disorders, too. These are treatment-related side effects that cannot be avoided in cancer treatment. For example, Cisplatin is known to indicate an anticancer action that suppresses the division of cancer cells by causing DNA damage of cancer cells. However, the patients who were treated with cisplatin suffer from experienced nausea [10,11] or side effects of various neurological symptoms [12] and kidney failure [12,13]. Likewise, there are also the adverse effects of radiotherapy. It is used with anticancer drugs or to reduce the size of preoperative cancer and the possibility of postoperative recurrence but there is also a high risk of side effects that cause brain metastasis [14], as in the case of brain radiation therapy. 

Beyond each symptomatic side effect of those treatments, those antineoplastic drugs have been shown to induce oxidative stress which can interfere with cell cycle progression and apoptotic pathways [15]. This can lower the effect of antineoplastic activity from chemotherapy and affect normal cells. Also, radiation therapy produces oxidative stress related to reactive oxygen species (ROS) signaling, deoxyribonucleic acid (DNA) damage response, membrane lipid peroxidation, mitochondrial damage, endoplasmic reticulum stress (ER stress) and autophagy [16]. During this cellular stress, p53 may play a pivotal role and it is supposed to ameliorate radiation-induced oxidative stress [17]. As a result, certain antioxidants appear to prevent some chemotherapy-induced side effects and there are also some plants that can get involved in some mechanisms from radiation-induced oxidative stress. 

### 1.2. Plants-Derived Drugs for Side-Effects of Cancer Therapies and Cachexia

Until now, research has been conducted on how to suppress the cancer therapeutics. However, including sequela of anti-cancer treatments, pharmacological clinical studies of cachexia have not been fully conducted. Considering patients’ treatment rates and quality of life, we should develop a way to handle them with minimal risks in the future. In spite of the complex mechanisms and symptoms related to oxidative stress, the efficacy of suppressing the side effects and cachexia is being studied with natural products including plant extracts. Therefore, those studies regarding antioxidant and anti-inflammation mechanisms of plant-derived drugs were reviewed in this study for the first time which manage side effects of cancer treatments and cachexia. 

## 2. Sequela of Surgery and Plant Extracts

### 2.1. Surgery

Surgery is the method of treatment to remove the neoplastic mass and nearby tissue [2]. The most common types of cancer surgeries that have been studied are liver cancer, pancreatic cancer, colon cancer and rectal cancer surgeries. In case of liver cancer, hepatectomy and liver transplantation are the most important approaches for achieving long-term survival especially for liver cancer stages Ιa, Ιb, ΙΙa and those with ≤3 tumors in stage Ⅱb and Ⅲa. [18]. For pancreatic cancer, there is a strong tendency to locally penetrate the surrounding nerves and lymph nodes to cut them extensively to where metastasis is expected. Similarly, the proper surgical principal for colon and rectal cancer is to resect the intestine at a sufficient distance from the tumor to the distal and proximal regions, along with affected lymph nodes. For this purpose, laparoscopic surgery has the advantage of fast recovery due to smaller incisions and less damage to surrounding organs during surgery. Despite the development of many surgical methods to reduce complications, there are large number of side-effects including pain, swelling, numbness and so forth [19,20]. It is still necessary to put a lot of effort into caring for the overall condition of the patient’s body after surgery.

### 2.2. Sequela of Surgery

Surgery is a general treatment for patients with solid tumors. However, cancer-related surgeries significantly increase cancer recurrence risk due to surgery-induced stress such as inflammation, ischemia-reperfusion injury, sympathetic nerve system activation and increased cytokine release [21]. Excisional surgery can also alter subsequent tumor growth and affect minimal residual disease (MRD), leading to perioperative tumor growth [22].

### 2.3. Plant-Derived Drugs and Sequela of Surgery

Cancer surgery can cause many side effects such as weight loss, fatigue, sickness and loss of appetite depending on the invaded location and size. Several natural products, such as epigallocatechin gallate (EGCG) [23], 2′,4′-Dihydroxy-6′-methoxy-3′,5′-dimethylchalcone [24], hyperforin and nemorosone [25], Daickenchuto [26,27] and so forth were reported to be effective against these side effects (Table 1). Stingl et al. suggested that epigallocatechin gallate (EGCG) extracted from green tea minimized the risk of metachronous adenomas of the colorectum with colorectal polyp removal surgery [23]. The 150 mg of EGCG was administered for 3 years to patients with polyp removal surgery. It showed more effective efficacy of prevention and treatment of colon cancer than the patients with only surgery. This result showed that the potential of EGCG induced interactions by the inhibition of major human cytochrome P450 isoenzymes. Huang et al. discovered that 2′,4′-Dihydroxy-6′-methoxy-3′,5′-dimethylchalcone had the modulatory effect on the 5-fluorouracil (5-FU) in resistant tumor hepatocellular xenograft with BEL-7402/5-FU cell line [24]. 2′,4′-Dihydroxy-6′-methoxy-3′,5′-dimethylchalcone, isolated from buds of *Cleistocalyx operculatus*, was injected at a dose of 20 or 40 mg/kg to nude mice for 16 days intraperitoneally and was measured every 2 days. As a result, combined therapy resulted in enhanced tumor apoptosis and reduced proliferative activities compared to 5-FU alone group [24]. Wolf et al. observed that nemorosone inhibited the cancer growth induced by pancreatic cancer xenografts surgery [25]. When nemorosone is injected at a dose of 50 mg/kg for 1 day, cytochrome P450 3A4 (CYP3A4)-independent oxidation products were increased. It means that nemorosone is a potential anti-cancer lead compound with good bioavailability, little side-effects and promising growth-inhibitory effects. The results showed the growth-inhibitory potential of nemorosone on pancreatic cancer xenografts in NMRI nu/nu mice and determined basic pharmacokinetic parameters. Katsuno et al. demonstrated that Daikenchuto accelerated the recovery of gastrointestinal function in patients undergoing open colectomy for sigmoid or rectosigmoid cancer [26]. Patients received either placebo or Daikenchuto (15.0 g/day, 5 g three times a day) from postoperative day 2 to day 8. As a result, Daikenchuto had a positive effect on the resolution of delayed gastric emptying while maintaining a good safety and tolerance profile. But it had a limited effect on the resolution of postoperative paralytic ileus after open surgery in patients with sigmoid or rectosigmoid cancer [26]. Yamada et al. reported that Daikenchuto was effective on the postoperative intestinal motility and anti-inflammation in most of the patients with sigmoid colon cancer surgery, especially in elderly patients [27]. Eighty-eight patients with right-side colon cancer were orally administered 7.5 g of Daikenchuto for 2 h and their C-reactive protein (CRP) was increased especially in elderly patients. This indicated that Daikenchuto is useful for patients with right-sided colon cancer and it is limited in elderly patient. Alleviation of post-surgical complications was reported in various experiment models. Potential of plant-derived drugs as a treatment after xenograft surgery was demonstrated in either in vivo or in vitro. Through combined therapy of 2′,4′-Dihydroxy-6′-methoxy-3′,5′-dimethylchalcone, isolated from buds of *Cleistocalyx operculatus*, with 5-FU, stimulation in tumor apoptosis and inhibition in proliferative activities were observed in BEL-7402/5-FU cell line and nude mice. In NMRI nu/nu mice model, regulated cancer growth was also reported through nemorosone administration [24,25]. Daikenchuto indicated efficacy on recovering gastrointestinal function in patients after surgery. Gastric emptying delay was improved in patients with sigmoid or rectosigmoid cancer surgery and postoperative intestinal motility was enhanced especially in elderly patients with sigmoid colon cancer surgery [26,27].

## 3. Sequela of Chemotherapy and Plant Extracts

### 3.1. Chemotherapy

Chemotherapy is the type of cancer therapy which is treatment of one or more cytotoxic and anti-tumor drugs such as alkylating agents [28], antimetabolites [29], mitotic inhibitors [30], topoisomerase inhibitors [31], corticosteroids [32] and so forth. It is commonly used in neoadjuvant, adjuvant, maintenance and palliation therapy depending on patient’ status. Anti-cancer drugs have evolved in a more effective manner over time. Chemotherapeutic agents have the principle of directly attacking cells that rapidly differentiate but this has the fatal disadvantage of attacking normal cells with rapid differentiation. Among normal cells, hematopoietic stem cells of the bone marrow, hair root cells, epithelial cells of the oral cavity and intestinal mucosa tend to proliferate rapidly and thus are affected by chemotherapy. This causes side effects such as leukocyte reduction, hair loss, vomiting and diarrhea. Targeted therapy [33], the second generation, executes cancer cells by suppressing the causes which are necessary for cancer cells to grow, not cancer cells. However, it has the disadvantage that it is less effective for cancer cells with drug resistance. Immunotherapy, the third generation, is manipulating immune checkpoints or pathways and chimeric antigen receptor T cells to brake off the immune system and allow it to attack cancer cells [34]. The mechanism of blocking immune checkpoints is about treating with anti-PD-L1 or anti-PD-1 that block the binding of PD-L1 ligand molecules of tumor cells to PD-1 molecules of T cells. For example, some of immune-cancer drugs are Ipilimumab, Nivolumab, Pembrolizumab, Atezolizumab and Avelumab. These immunotherapies commonly induce fatigue, diarrhea, fever, shortness of breath, rash, nausea, vomiting, itching, weight loss, insomnia and decreased appetite. These side effects may be represented as the type of inflammation in organ such as lung, liver, colon or kidney and so forth. Nowadays, immunotherapy extends beyond immune checkpoint therapy by managing chimeric antigen receptor T cells and chimeric monoclonal antibodies that target malignant cells to be destroyed effectively [35].

### 3.2. Sequela of Chemotherapy

Chemotherapy-induced side effects remain as major clinical limitations even though chemotherapy is widely used to treat cancer. General side effects caused by chemotherapies include cisplatin-induced emesis [36,37,38], methotrexate-induced hepatotoxicity [39,40], benzo (a) pyrene-induced immune disability [41] and cisplatin-induced skeletal muscle dysfunction [42]. Anticancer drugs included in our research are carboplatin, cisplatin, cyclophosphamide, docetaxel, doxorubicin, 5-fluorouracil, methotrexate, oxaliplatin, taxane, TS-1 and vincristine, which often result in adverse effects. Carboplatin induces myelosuppression and causes intestinal mucositis through alternations in intestinal microbial components [43,44]. Nephrotoxicity and ototoxicity can occur as side effects of cisplatin [45]. Cyclophosphamide leads to depletion of regulatory T cells, followed by the induction of myeloid-derived suppressor cells (MDSC) [46] and also increases ROS generation [47]. Hyper-eosinophilia is frequent under docetaxel treatment, which leads to hypersensitivity reactions [48]. Doxorubicin causes cardiotoxicity by altering the cellular response to ROS [49]. 5-Fluorouracil leads to oral mucositis, intestinal mucositis with disturbed gut microbiota and cardiotoxicity [50,51,52]. Methotrexate induces cognitive impairment and oral ulcers [53,54]. Oxaliplatin can bring about neurotoxicity through alternation of blood brain barrier (BBB), thrombocytopenia and intestinal dysfunction [55,56,57]. Taxane-based chemotherapy induces alopecia as well as nail and cutaneous changes [58] and TS-1 can induce myelosuppression and leukopenia [59]. The main side effect of vincristine includes peripheral neuropathy and inappropriate secretion of antidiuretic hormone (ADH) [60,61]. 

### 3.3. Plant-Derived Drugs and Sequela of Chemotherapy

Many researchers showed that natural products are effective for reducing the side effects induced by several chemotherapies via regulation of related factors (Table 2 and Figure 1). Chen et al. reported that Curcumin, derived from *Curcuma longa* Linn, attenuated carboplatin-induced myelosuppression by activating the DNA repair pathway in bone marrow cells [62]. Up-regulation of breast cancer gene (BRCA)1, BRCA2 and excision repair cross-complementing (ERCC)1 expression in bone marrow were identified following the treatment of Curcumin 5 mM for 15 days in T241-bearing mice. These findings demonstrate that Curcumin attenuates the carboplatin-induced myelosuppression. Standardized methanolic and *n*-butanolic fractions of *Bacopa monnieri* suppressed the cisplatin-induced reproducible emesis of pigeon without lethality [63]. The pigeons were administered with 10, 20, 40 mg/kg of fractions for up to 24 h after treated with cisplatin. As a result, *Bacopa monnieri* attenuated the vomiting of pigeon via decreased dopamine, 5-hydroxytryptamine (5-HT) (serotonin) and 5-hydroxyindoleacetic acid (5-HIAA). Zhou et al. demonstrated that Cepharanthine hydrochloride can be used as a potential clinical treatment in esophageal squamous cell carcinoma [64]. Activation of cleaved poly ADP ribose polymerase (c-PARP), c-caspase-3, -8, -9, tumor necrosis factor receptor 1 (TNFR1), p- Jun *N*-terminal kinase (JNK), cytochrome C and the reduction of B-cell lymphoma 2 (Bcl-2) were observed following treatment of 10 mg/kg of Cepharanthine hydrochloride for 13 days in Ecal109 cells-inoculated BALB/c nude mice. These findings demonstrate that Cepharanthine hydrochloride increases the anti-cancer effect of cisplatin and reduces the side effects on the microbes and intestinal mucosal immunity induced by cisplatin. Pomegranate seed extract isolated from *Punica granatum* demonstrated that cisplatin-induced acute nephrotoxicity and hepatotoxicity were attenuated [65]. Simultaneously, it upregulated glutathione, glutathione S-transferase, peroxidase, superoxide dismutase and decreased lipid peroxidation, malondialdehyde (MDA), caspase-3. Such results were identified when its dosage was 300 mg/kg for 15 days in SD rats. Pan et al. showed that Theaflavin-3-gallate (TF2a) and theaflavin-3′-gallate (TF2b), the theaflavin monomers in black tea, exhibited a potent growth inhibitory effect on cisplatin-resistant ovarian cancer A2780/CP70 cells [66]. With the treatment of 5, 10 and 20 µM of TF2a and TF2b for 24 h, the elevation of caspase-3, -7, PARP, p21, p38, p53 and the down-regulation of cyclin-dependent kinase2 (CDK2), CDK4 and cyclin E1 were observed. These findings indicate that the inhibitory effect of TF2a and TF2b treated platinum-resistant ovarian cancer by inducing apoptosis and G1 cell cycle arrest. Rajkumar et al. reported that the administration of avocado methanol extract (100, 200, 300 mg/kg) from *Persea Americana* Mill for 70 h to human lymphocyte reduced its genotoxicity from cyclophosphamide [67]. It decreased chromosomal aberrations and neutralized numerical aberrations. It also reduced structural aberrations, acrocentric association and premature centromeric division. Murali et al. reported that *Curculigo orchioides* methanolic extract, originated from *Curculigo orchioides* Gaertn, enhanced the anticancer properties and ameliorated the toxic side effects of cyclophosphamide [68]. Elevation of α-esterase, interleukin (IL)-2, granulocyte monocyte colony-stimulating factor (GM-CSF), interferon (IFN)-γ, glutathione (GSH) and the downregulation of tumor necrosis factor (TNF)-α, lipid peroxidation (LPO), alkaline phosphatase (ALP), glutamate pyruvate transaminase (GPT) were identified with the treatment of *Curculigo orchioides* methanolic extract 25, 50, 100, 200 mg/kg for 14 days in Balb/c mice and Swiss albino mice. These results indicate the *Curculigo orchioides* methanolic extract as the possible treatment for myelosuppression and cyclophosphamide-induced oxidative stress. The chemoprotective effect of *Decalepis hamiltonii* was demonstrated in cyclophosphamide-induced toxicity model of BALB’s mice by injecting its methanolic extract (0.5 mg) for 10 days [69]. As a result, this increased white blood cell (WBC) count, bone marrow cellularity, α-esterase positive cells, weights of spleen and lungs and reversed serum GSH levels. It also decreased the levels of serum glutamate oxaloacetate transaminase (SGOT), serum glutamate pyruvate transaminase (SGPT), urea, creatinine, blood urea nitrogen (BUN) and reduced superoxide dismutase (SOD) activity. Zarei et al. identified that the roots of *Decalepis hamiltonii* Wight & Arn protected brain from oxidative stress induced by cyclophosphamide and ameliorated the toxic side effects of cancer drugs [70]. Up-regulation of GSH, SOD, catalase (CAT), glutathione peroxidase (GPx), glutathione reductase (GR), glutathione-*S*-transferase (GST) and decrease of reactive oxygen species (ROS) were observed in Swiss albino male mice. They were pretreated with *Decalepis hamiltonii* Wight & Arn 50 and 100 mg/kg for 10 consecutive days followed by an injection with cyclophosphamide 25 mg/kg for 10 days. These results indicate that the roots of *Decalepis hamiltonii* Wight & Arn can be a promising nutraceutical as a supplement in cancer chemotherapy. Wang et al. demonstrated that green tea and quercetin enhanced the therapeutic effect of docetaxel in castration-resistant prostate cancer cells such as LAPC-4-AI and PC-3 [71]. With the treatment of green tea 40 μM for 48 h and quercetin 5 μM for 48 h, the increase of Bcl-2 associated X (Bax)/Bcl-2 and the decrease of phosphatidylinositol 3-kinase (PI3K)/protein kinase B (Akt), signal transducer and activator of transcription 3 (STAT3), multidrug resistance-associated protein 1 (MRP1) and CD44+/CD24− stem-like LAPC-4-AI cells were detected. Mendanha et al. showed that administration of *Byrsonima verbascifolia* protected doxorubicin-induced damage [72]. The experiments were carried out at a dose of 25, 50, 100 mg/mL for 2 days to somatic cells of Drosophila melanogaster. The effect could be detected when mutant spots in standard (ST) and high bioactivation (HB) cross were reduced in a dose-dependent way. Hou et al. reported that ginsenoside Rh2 ameliorated doxorubicin-induced senescence bystander effect in MDA-MB-231 breast carcinoma cells [73]. Treatment of ginsenoside Rh2 20 μg/mL for 2 days decreased beta-catenin, vimentin, caspase 3/7, monocyte chemoattractant protein-1 (MCP-1), CXCL1, IL-6, IL-8, p- mitogen-activated protein kinase kinase 1 (MEK1), p-p38, p-STAT3, phosphor-nuclear factor kappa-light-chain-enhancer of activated B cells (p-NF-κB) p65 in MDA-MB-231 cells and decreased caspase 3/7, MCP-1, CXCL1, IL-6, IL-8, p38, STAT3 in MCF-10A cells. These results indicate that ginsenoside Rh2 attenuates doxorubicin-induced cellular senescence and SASP. Cohen Z et al. showed that LCS101 treatment at doses of 1, 2, 3 mg/mL for 24, 48, 72 h induced the anti-proliferative effect on breast, colorectal, prostate cancer cells as necrosis-like features [74]. Also, it selectively protected non-tumorigenic cells from doxorubicin and 5-Fluorouracil by reducing PARP-1 cleavage. These results suggest that LCS101 provides selective cytotoxic effect to cancer cells and protective effect to normal cells exposed to doxorubicin and 5-Fluorouracil. Freitas reported that laticifer proteins, isolated from *Calotropis procera*, attenuated 5-fluorouracil-induced oral mucositis in golden hamsters [75]. The observing groups were injected 0.25, 1, 5, 25 mg/kg for 24 h before and after mechanical trauma. This treatment shows downregulated expression of cyclooxygenase-2 (COX-2), inducible nitric oxide synthase (iNOS), TNF-α, IL-1β and reduced myeloperoxidase (MPO) activity. Xi et al. reported that the intragastric administration of Ciji Hua’ai Baosheng Granule Formula (CHBGF) prolonged the survival time of H22 hepatoma carcinoma cells-injected mice [76]. The mice were administered with CHBGF (16, 32, 64 g/kg) once a day and peritoneal injection of 5-Fluorouracil (25 mg/kg) once every other day for 21 days. As a result, the CHBGF decreased the growth of tumor, WBC and platelet, while increasing the red blood cell (RBC) and hemoglobin. Omid et al. identified the healing effect of *Echinacea angustifolia* de Candolle in 5-flurouracil-induced oral mucositis in golden hamster [77]. Increase of SOD and decrease of MPO were detected with the treatment of hydroalcoholic extract of *Echinacea angustifolia* de Candolle 3000 mg/kg for 5 days. These results revealed that *Echinacea angustifolia* de Candolle is the appropriate drug choice for treating oral mucositis. Kato et al. showed that Saireito, the traditional Japanese herbal (Kampo) medicine, prevented 5-fluorouracil induced intestinal mucositis, body weight loss, diarrhea during cancer therapy [78]. Following the treatment of Saireito 100, 300, 1000 mg/kg for 6 days in C57BL/6 mice, the reduction of caspase-3, TNF-α, IL-1β was identified. These findings demonstrated that Saireito, by inhibiting cytokine-mediated apoptosis in intestinal crypt cells, may be clinically useful for the prevention of intestinal mucositis during cancer chemotherapy. Kim et al. demonstrated that the dried root of *Salvia miltiorrhiza* Bunge can be used as a potential clinical treatment in oral mucositis caused by 5-flurouracil [79]. Elevation of 2-diphenyl-1-picrylhydrazyl (DPPH) and downregulation of NF-κB, caspase-3, ROS, TNF-α, IL-1β were examined following the treatment of *Salvia miltiorrhiza* 100, 500, 1000 mg/kg for 10 days. Famurewa et al. demonstrated that Zobo, the *Hibiscus sabdariffa* Linn extract, abrogated hepatic damage by targeting oxidative hepatotoxicity [80]. Elevation of SOD, CAT and GPx and suppression of MDA were observed following the treatment of Zobo 10 mL/kg for 14 days in Albino Wistar rats. These results suggest that Zobo can be a superior protection against Methotrexate-induced oxidative hepatotoxicity by improving antioxidant defense systems. Yue et al. identified that Huachansu, the aqueous extract from toad skin, derived from *Bufo bufo gargarizans* Cantor, can have therapeutic potential for the treatment and prevention of chemotherapy-induced peripheral neuropathic pain [81]. Injection of Huachansu 2.5 g/kg for 21 days induced elevation of transient receptor potential vanilloid 4 (TRPV4) and suppressed TRPV1 up-regulation and spinal astrocyte activation. These findings indicate that Huachansu prevents oxaliplatin-induced peripheral neuropathic pain. Cinci et al. suggested the *Hypericum perforatum* Linn extract as the novel therapeutic strategy for neuropathy caused by oxaliplatin [82]. The decrease of caspase-3 was observed with the treatment of 5, 50, 250 μg/mL for 4 and 8 h in rat astrocytes and HT-29 cells. This result demonstrates a significant antioxidant effect of the *H. perforatum* Linn extract. Riccio et al. suggested that Annurca apple polyphenolic extract originated from *Malus pumila* Miller cv. Annurca protected the murine hair follicles from taxane induced dystrophy [83]. Elevation of prostaglandins F2α (PGF2α) and suppression of pentose phosphate pathway (PPP) were detected in following treatment of Annurca Apple Polyphenolic extract at a dose of 400 mg/L for 7 days in wild-type C57BL/6 mice. Juzentaihoto treatment activated hematopoiesis in the TS-1 administered Balb/c mice model [59]. When it was treated with 1 g/kg orally for 3, 5, 7 days, WBC count and CD34+ bone marrow cells (BMC) ratio were improved on the last day. Also, this result showed C-terminal fragment of albumin was a candidate biomarker myelosuppression in TS-1 therapy. Park et al. identified that orally administered *Ginkgo biloba* showed dose-dependent anti-hyperalgesic effect [84]. In vincristine-induced peripheral neuropathy rat model, which was administered with *Ginkgo biloba* (50, 100, 150 mg/kg) for 30, 60, 90, 120, 150, 180 min, the mechanical withdrawal threshold was increased and withdrawal frequency to cold stimuli was significantly reduced.

Plant-derived drugs showed exceptional efficacy on organ toxicity, neurotoxicity, genotoxicity, mucositis and impaired hematopoiesis. Organ toxicity induced by cisplatin resulted emesis, acute nephrotoxicity, hepatotoxicity and cisplatin-resistant ovarian cancer and those sequela were alleviated [63,65,66]. Cepharanthine hydrochloride reduced the side effects on the microbes and intestinal mucosal immunity induced by cisplatin [64]. *Decalepis hamiltonii* alleviated the toxicity and oxidative stress in brain [69,70]. *Hibiscus sabdariffa* Linn extract demonstrated protection against methotrexate-induced oxidative hepatotoxicity [80]. Also, neurotoxicity is a common side effect of oxaliplatin and vincristine. Related side effects were effectively relieved by plant extracts. Huachansu prevented peripheral neuropathic pain [81] and *Hypericum perforatum* Linn extract alleviated neuropathy [82], which were caused by oxaliplatin. Orally administered *Ginkgo biloba* showed dose-dependent anti-hyperalgesic effect on vincristine-induced peripheral neuropathy [84]. Alleviation of genotoxicity were observed. Avocado methanol extract decreased cyclophosphamide-induced chromosomal aberrations [67] and ginsenoside Rh2 attenuated cellular senescence induced by doxorubicin [73]. Oral mucositis [75,77,79] and intestinal mucositis [78,85] induced by 5-fluorouracil were also effectively ameliorated. Improvement of hematopoiesis were shown. *Curculigo orchioides* methanolic extract ameliorated cyclophosphamide-induced myelosuppression [68]. Juzentaihoto treatment activated hematopoiesis in the TS-1 administered Balb/c mice model [59].

## 4. Sequela of Radiotherapy and Plant Extracts

### 4.1. Radiotherapy

Radiotherapy is the treatment of cancer which uses high-energy radiation such as x-rays and gamma rays. It is usually used in addition to surgery and chemotherapy [86]. There are several different types of radiation therapy such as external beam radiation, particle therapy and internal radiation therapy. External beam radiation uses x-rays and particle therapy uses protons or electrons that both come from external to the body. This can cause gradual side effects like fatigue and some skin irritation. Each person may have a different treatment regimen depending on their individual specificity and the type of cancer but every single tissue in the body has a different tolerance to radiation therapy. These are why radiation therapy induces side effects and it could be relieved by preventing dehydration and managing skin moisture.

### 4.2. Sequela of Radiotherapy

Radiotherapy is considered as frontline cancer treatment but many complications have been reported. Radiotherapy can be a causative factor of immune deterioration and oxidative stress. By disrupting the function of myeloid-derived suppressor cells (MDSCs), which are major regulators of immune response in cancer, radiotherapy has adverse effect on our immune system [87]. Normal tissues receiving irradiation may have significant levels of clustered DNA damage, which leads to mutations, chromosomal aberrations and ultimately secondary cancers [88]. Oral and gastrointestinal mucositis, esophagitis and dermatitis are frequent sequela of radiotherapy [89,90,91,92,93]. Radiation can also induce toxicity in organs such as heart, kidney and liver. Patients receiving radiation at the thoracic zone often show cardiotoxicity including pericardial disease and cardiovascular disease [94,95]. Urinary symptoms such as dysuria, hematuria, incontinence and frequency are also common sequela [96]. Radiation-induced liver injury is an unusual but relevant complication [97]. Regarding the complications of radiotherapy, effects of natural products on relieving injury and toxicity in organs were observed.

### 4.3. Plant-Derived Drugs and Sequela of Radiotherapy

Recently, numerous researches have reported plant-derived materials that can mitigate radiation-induced side effects (Table 3). The mechanisms of drugs were elucidated in Figure 2. Lee et al. reported that Bojungikki-tang, composed of *Panax ginseng* C. A. Meyer, *Atractylodes macrocephala* Koidz., *Astragalus membranaceus* Bunge, *Angelicae gigantis* Radix, *Citrus aurantium* Linne, *Ziziphus jujuba* var. inermis, *Bupleurum falcatum* Linne, *Glycyrrhiza uralensis* Fisch., *Zingiber officinale* Roscoe, *Cimicifuga heracleifolia* KOM., showed the elevation of CD19+ B cell in peripheral blood [98]. This result suggests that the oral administration of Bojungikki-tang for 4 weeks restores the decreased B cells and recovers the immune deterioration induced by localized radiotherapy. Hamilton et al. reported that cranberry, which is derived from *Vaccinium macrocarpon* Ait., alleviated the radiation-induced cystitis in prostate cancer patients [99]. The patients were treated with a capsule containing 72 mg of cranberry once a day for 2 weeks. As a result, cranberry treated patients showed less pain and burning, better control, a stronger urine stream and less leaking. In another study about cranberry, it prevented the bladder mucosa damage during radiotherapy for prostate carcinoma [100]. One enteric-coated tablet per day containing 200 mg of a highly standardized cranberry extract mitigated lower urinary tract infections and urinary symptoms such as dysuria, nocturia, urinary frequency, urgency when they were treated for 7 weeks. Ginger, extracted from *Zingiber officinale*, upregulated total antioxidant capacity (TAC) and downregulated 8-hydroxy-2′-deoxyguanosine (8-OhdG), CRP, cystatin C/creatinine ratio when pretreated in Wistar rat model for 10 days at doses of 50 mg/kg [101]. This indicated that ginger suppresses oxidative DNA damage, inflammatory reactions, histological and biochemical alterations in kidney tissues after γ-ray exposure. Ji et al. reported that increased levels of heme oxygenase-1 (HO-1) and NAD(P)H:quinone oxidoreductase 1 (NQO-1) with decreased levels of ROS demonstrated that ginger mitigates ionizing radiation-induced cell injury in human mesenchymal stem cells [102]. Activation of cytoprotective genes encoding for HO-1 and NQO-1 and nuclear factor erythroid 2-related factor 2 (Nrf2) protective response were observed at a dose of 1, 10, 100, 1000 μg/mL for 24, 48, 72 h. In addition, oral pretreatment of ginger essential oil on Balb/C mice at a dose of 100 and 500 mg/kg restored decreased hematological and immunological parameters [103]. Increased levels of SOD, catalase, GPx, GSH, micronucleated polychromatic erythrocytes (MNPCE), micro nucleated normochromatic erythrocytes (MNNCE), polychromatic erythrocytes/normochromatic erythrocytes ratio (P/N ratio) suggested that ginger inhibits γ-irradiation induced oxidative stress and clastogenic damage. Hangeshashinto is composed of *Pinellia ternata*, *Scutellaria baicalensis*, *Zingiber officinale* Roscoe, *Glycyrrhiza uralensis* Fisch., *Zizyphus jujuba* Mill., *Panax ginseng* C. A. Meyer, *Coptis chinensis* Franch [104]. 2% Hangeshashinto injection to Syrian golden hamsters for 28 days suppressed COX-2 expression and neutrophil infiltration, which led to mitigation of radiation-induced mucositis. Balaji et al. reported that pure natural honey alleviated radiation mucositis when topically applicated at a dose of 20 mL, three times per day, for 7 weeks [105]. Following treatment prevented more debilitating grade IV mucositis, delayed the onset of mucositis, accelerated oral mucosa entering a healing phase and completely recovered patients without any symptoms. *Panax ginseng* water extract prevented radiation-induced liver injury at a dose of 25, 50, 100 mg/kg when orally administered to a C57BL/6N mice model for 4 days [106]. The treatment of *Panax ginseng* water extract affected body and liver weights and hematotoxicity. Also, activation of WBC, TAC, GSH, glutathione reductase (GSH-Rd), SOD, catalase, Bcl-2, B-cell lymphoma-extra-large (Bcl-xL) and inhibition of 4-hydroxynonenal (4-HNE), ROS, hepatic methylenedioxyamphetamine (MDA), hepatic triglyceride, alaninetransaminase (ALT), ALP, TNF-α, IL-6, p53, Bax were observed. PHY906, which consists of *Scutellaria baicalensis*, *Glycyrrhiza uralensis* Fisch., *Paeonia lactiflora*, *Ziziphus jujuba* var. inermis, mitigated the toxicity and promoted rapid recovery from fractionated abdominal irradiation [107]. However, the following treatment could not alter the growth or radiation responses of the tumors. PHY906 was given with a dose of 500 mg/kg/fraction for 4 days, which was administered to BALB/c Rw mice and cell culture were performed using EMT6 mouse mammary tumor cells. Ghassemi et al. showed that ethanolic extract of propolis delayed the first sign of mucositis, decreased the severity of mucositis especially in lip and tongue by reduction of the infiltration of inflammatory cells [108]. Such results were observed at doses of 100, 200 mg/kg for 10 days in Wistar rat model and efficacy was dose-dependent manner. However, the optimal dose of ethanolic extract of propolis is not yet known, thus further study is needed. Shenqi Fuzheng, containing *Codonopsis pilosula* and *Astragalus membranaceus* Bunge, suppressed the expression of horseradish peroxidase (HRP), TNF-α, IL-1β, NF-κB, C-terminal fragment of p53-induced protein segments with a death domain (PIDD-C), the twice-cleaved fragment of p53-induced protein with a death domain (PIDD-CC), p65 [109]. This result suggests that the following treatment alleviates the irradiation-induced BBB permeability injury, inhibits NF-κB activation, decreases the number of activated microglia and apoptotic cells when administered intraperitoneally at a dose of 20 mL/kg/d for 28 days. Yang et al. reported that Zerumbone, originated from *Zingiber zerumbet* Smith, prevented Ultraviolet A (UVA)-induced skin damage and photoaging via increased nuclear localization of Nrf2 and Nrf2-dependent antioxidant genes [110]. Administrating 55 and 110 μg/day for 24 h to nude mice, which is more susceptible to photoaging induced by UVA exposure, increased Bcl-2, Nrf2, HO-1, γ-glutamyl cysteine ligase (γ-GCLC), GSH, p38 mitogen activated protein kinase (MAPK), PI3K and decreased lactate dehydrogenase (LDH), ROS, Bax. Compound Zhuye Shigao Granule, consisting Lophatherum gracile Brongn., Gypsum, *Panax ginseng* C. A. Meyer, *Liriope platyphylla*, Pinellia ternate (Thunb.) Breit., *Glycyrrhiza uralensis* Fisch., *Rabdosia serra* (Maxim.) Hara, Hedyotis diffusa Willd., *Scutellaria barbata* D. Don, Coix lacryma-jobi, *Curcuma longa* Linne, decreased the incidence, grade, duration, damage of acute radiation-induced esophagitis and delayed the time of occurrence [111]. Following treatment were given 12 mg orally for 4 weeks. Zingerone, which is derived from *Zingiber officinale*, expressed antioxidant, anti-inflammatory and antiapoptotic effect against cisplatin- or γ-radiation-induced cardiotoxicity [29]. Upregulation of GSH, CAT, electron transport chain (ETC) complex I, II, IV and downregulation of cardiac troponin T (cTnT), LDH, creatine kinase-MB (CK-MB), myocardial malondialdehyde (MDA), TNF-α, MPO, caspase-3 were observed at a dose of 25 mg/kg for 21 days. Another study of Zingerone demonstrated that Zingerone protected keratinocyte stem cells from UVB-induced damage at a dose of 10, 20, 100 μM for 24 h [112]. Following treatment induced the activation of proliferating cell nuclear antigen (PCNA), vascular endothelial growth factor (VEGF), telomerase reverse transcriptase (TERT), histone deacetylase 1 (HDAC1), DNA (cytosine-5)-methyltransferase 1 (DNMT1), while inhibiting TNF-α, IL-1β, IL-6, p21, p42/44 MAPK, p38 MAPK. This result indicates that Zingerone inhibits the Ultraviolet B (UVB)-mediated production of pro-inflammatory cytokines, affects expression of cell cycle arrest and cell survival-related genes, attenuates UVB-induced cell damage. Meimeti et al. suggested that gel containing Pinus halepensis bark aqueous extract and ointment containing olive oil extract of the marine isopod Ceratothoa oestroides mitigated skin injury induced by X-ray irradiation [113]. When 5% Pinus halepensis bark aqueous extract and 10% ointment containing olive oil extract of the marine isopod Ceratothoa oestroides were topically applied for 60 days, alleviation of skin redness and hydration was observed.

Efficacy of plant-derived drugs was apparent in organ toxicity, inflammation, skin damage after radiation therapy. Organ toxicity such as radiation-induced urinary symptoms in prostate cancer patients and liver injury was prevented [99,100,106]. Herbal formula called Shenqi Fuzheng mitigated the irradiation-induced brain injury via inhibition of the NF-κB signaling pathway [109]. Zingerone expressed antioxidant, anti-inflammatory and antiapoptotic effect against cardiotoxicity [29]. Anti-inflammatory effects on mucositis and esophagitis were also suggested. Plant derived drugs had prophylactic and delaying efficacy in radiation mucositis when administered orally and topically [104,105,108]. Decrease in incidence, grade, duration, damage and occurrence of acute esophagitis were observed [111]. Skin damage caused by radiation was reduced. Zerumbone increased nuclear localization of Nrf2 and Nrf2-dependent antioxidant genes [110] and Zingerone protected keratinocyte stem cells from UVB [112]. Topically applied *Pinus halepensis* gel and *Ceratothoa oestroides* ointment also mitigated skin redness and hydration [113]. In addition, some natural products showed consistent efficacy in radiation-induced sequela even in different extraction methods. For example, Ginger hydro-alcoholic extract suppressed oxidative DNA damage and inflammatory reactions, ginger oleoresin showed mitigation of radiation-induced cell injury and ginger essential oil alleviated oxidative stress and clastogenic damage [101,102,103].

## 5. Cachexia and Plant Extracts

### 5.1. Cachexia

Patients commonly undergo not only treatment-related symptoms but also cancer-induced symptoms which can be categorized as cancer cachexia. In cancer patients, cachexia is characterized by systemic inflammation associating with decreased caloric intake, anorexia, decreased muscle strength and fatigue [114,115]. Cancer cachexia has been regarded as non-curable symptom and has devastating impact on patient’s quality of life and survival. Nevertheless, the severity tends to be underestimated [116,117]. The energy wasting has been attributed to inefficient adenosine triphosphate (ATP) production due to mitochondrial dysfunction. Dysfunctional mitochondria produce high levels of ROS causing oxidative damage to lipids and proteins with enhanced inflammation [118]. It is related to ER stress induced unfolded protein response (UPR) pathways and it causes catabolic conditions [119]. This may lead to myofibrillar protein breakdown, increased lipolysis, insulin resistance, elevated energy expenditure, reduced food intake and psychological factors related to quality of life [120]. These also do not act as a single variable but are related to the whole. This is why clinical research on cachexia treatment has been insufficient. 

### 5.2. Clinical Difficulties in Treatment of Cachexia

Current therapies focus on palliation of symptoms rather than cure. However, therapies such as simple improvement of nutritional intake are not enough considering that mechanisms of cancer cachexia are complex and multifactorial [121]. Numerous cytokines, hormones, metabolism and neurologic factors involve in the etiology of cancer cachexia. TNF-α, IL-6 and IL-1 mimic leptin signaling and suppress orexigenic ghrelin and neuropeptide (NPY) signaling, which result in weight loss and anorexia [122]. Upregulation of myostatin, an extracellular cytokine regulating hypertrophy, was observed in the pathogenesis of muscle wasting during cachexia [123]. Fatigue is induced by serotonin dysregulation, hypothalamic-pituitary-adrenal (HPA) axis disturbance, alteration in circadian rhythms, ATP dysregulation, vagal afferent nerve activation and cytokine dysregulation [124]. Treatment should be developed based on understanding of molecular mechanisms that induce cachectic symptoms. Absence of established guideline and insufficient clinical evidence for cancer cachexia treatment are another difficulty. Appetite stimulants, currently used to supplement calories and improve anorexia, do not always lead to weight gain. Agents affecting cachectic mediators or signaling pathways such as EPA, β-Hydroxy-β-methylbutyrate (HMB), Thalidomide, NSAIDs are also being used but further studies are required to confirm their efficacies [125]. Thus, the further research and clinical practice of cancer cachexia are needed to promote antioxidant and therapeutic effect.

### 5.3. Plant-Derived Drugs and Cachexia

There have been significant advances in cancer therapies. In spite of these progress, half of all patients with cancer commonly undergo cachexia which is associated with quality of life, poor response to cancer therapy and finally survival [116]. The symptoms include weight loss, anorexia, muscle atrophy, fatigue, anemia and so forth. Some compounds and extracts of plants were observed to decrease these symptoms (Table 4) and their mechanisms were organized in Figure 3. Li, B et al. elucidated that Bicalin, which is isolated from *Scutellaria baicalensis,* ameliorated the anorexia, weight loss and muscle atrophy [126]. CT26 adenocarcinoma inoculated male BALB/c mice was administered 50, 150 mg/kg of Bicalin by intraperitoneal injection for 15 days. These observing groups were demonstrated that p-p65/GAPDH band intensity was decreased and inhibitor of nuclear factor kappa B (IκBα)/GAPDH expression was elevated. This result suggests that Bicalin inhibits NF-κB activation which leads muscle wasting. Kim et al. observed that water extract of *Citrus unshiu* Markovich ameliorated weight loss, muscle wasting and Hb loss when its dosage was 150 and 500 mg/kg for 17 days [127]. The treatment not only decreased the levels of p-p38, extracellular signal-regulated kinase (ERK), JNK, IκBα and STAT3 but also mitigated lipopolysaccharide (LPS)-induced NO, iNOS expression and pro-inflammatory cytokine production. Also, it alleviated the CT-26-mediated C2C12 myotube wasting and changed myosin heavy chain (MyH), Akt phosphorylation and p65 phosphorylation. Curcumin and green tea extract, extracted from *Curcuma longa* and *Camellia sinensis* respectively, ameliorated weight loss and muscle wasting [128]. This treatment was implemented on C2C12 myotubes for 24 h as its dosage of 10 μg/mL. Such results were shown when TNF-α and proteolysis-inducing factor (PIF) was combined with not only curcumin and green tea extract but eicosapentaenoic acid (EPA). Rikkunshito was demonstrated to ameliorate anorexia and weight loss by upregulating hypothalamic orexigenic neuropeptide Y (NPY) and decreasing thyrotropin-releasing hormone (TRH) in paraventricular nucleus (PVN) [129]. However, Rikkunshito treatment did not affect the elevated plasma ghrelin levels and growth hormone secretagogue receptor (GHS-R) gene expression. Such results showed when 1 g/kg/day of Rikkunshito was orally administered twice daily for 7 days to 85As2 cells inoculated six-week-old male F344/NJcl-rnu/rnu rats. Shen et al. also reported that the oral administration of SiBaoChongCao (1, 2 g/kg for 20 days) which is isolated from *Cordyceps sinensis* ameliorated weight loss and muscle atrophy with shrinking adipocyte cell [130]. The muscle atrophy was alleviated by decreasing myosin heavy chain (MHC), myogenic differentiation antigen (MyoD), myogenic regulatory factors (MyoG), target of rapamycin kinase complex 1 (TORC1) and peroxisome proliferator-activated receptor gamma coactivator 1α (PGC-α) and upregulating AKT and mammalian target of rapamycin (mTOR) pathway. This result suggests that it could facilitate myoblast protein synthesis in muscle tissue not regardless of blocking protein degradation. Choi et al. observed that Sipjeondaebo-tang ameliorated cancer-induced anemia and anorexia including weight loss and muscle wasting in CT-26 tumor-bearing mice [131]. In this process, Sipjeongdaebo-tang suppressed IL-6 and monocyte chemoattractant protein-1 (MCP-1) but not TNF-α. It also affected anorexia by regulating glucagon like peptide-1 (GLP-1) and peptide YY (PYY) but not ghrelin and leptin. Finally, it increased the levels of RBC, Hb and hematocrits (HCT), which means treating anemia. Such results were shown when its dosage was 6.784, 67.84 and 678.4 mg/kg for 21 days. Kim et al. reported that oral administration of Soshio-tang (50, 100 mg/kg for 18 days) was identified to alleviate weight loss, muscle wasting and improve appetite [132]. As a part of the result, Soshio-tang strongly prevented LPS-induced p38, IκBα, IKKαβ and STAT3. However, the levels of ERK and JNK were not because this treatment inhibited p65 nuclear translocation, which can block NF-κB activation and attenuated muscle atrophy. Zhuang et al. demonstrated that Zhimu and Huangbai herb pair which was isolated from *Anemarrhena asphodeloides* and *Phellodendron amurense* ameliorated body weight loss and muscle protein catabolism [133]. This effect was carried out by activating insulin-like growth factor 1 (IGF-1)/Akt signal and increasing the expression of microtubule-associated protein 1A/1B-light chain 3 (LC3B) and sirtuin1 (SIRT1). Such results were shown when 104 mg/kg of Zhimu and Huangbai her pair was administered to colon-26 adenocarcinoma inoculated Male C57BL/6 mice for 18 days.

Common symptoms of cancer cachexia, including anorexia, weight loss and muscle atrophy, were ameliorated by plant-derived drugs through various mechanisms. Bicalin inhibited NF-κB activation [126], *Citrus unshiu* extract suppressed the production of pro-cachectic cytokines [127] and curcumin and green tea enhanced protein degradation through TNF-α and PIF [128]. Effectiveness of plant-derived drugs was prominent in herbal formulas or herbal pairs. It can be assumed that such tendency is due to complex and multifactorial characteristic of cachexia. Rikkunshito attenuated anorexia and weight loss by upregulating hypothalamic orexigenic NPY and decreasing TRH [129]. SiBaoChongCao facilitated myoblast protein synthesis in muscle tissue, preventing weight loss and muscle atrophy with shrinking adipocyte cells [130]. Sipjeondaebo-tang ameliorated cancer-induced anemia and anorexia [131]. Soshio-tang inhibited p65 nuclear translocation, which can block NF-κB activation [132]. Zhimu and Huangbai herb pair activated IGF-1/Akt signal, therefore preventing body weight loss and muscle protein catabolism [133].

## 6. Clinical Trials of Plant-Derived Drugs against Sequela of Cancer Therapies and Cachexia

Many clinical trials showed that plant-derived drugs reduce side effects of surgery, chemotherapy, radiotherapy and relieve symptoms of cachexia in USA, Japan, Australia, Iran and so forth. (Table 5). Daikenchuto, an herbal medicine of Japan, improves abdominal symptoms by accelerating bowel motility. Thus, it is widely used for prophylactic and therapeutic measures in patients with post-operative ileus. It consists of *Zingiber officinale* Roscoe, *Zanthoxylum piperitum* De Candolle, *Panax ginseng* Carl Anton Meyer and maltose. A randomized, double-blind, multicenter, placebo-controlled study showed that Daikenchuto improved gastrointestinal dysfunction in patients undergoing colectomy. One-hundred-seventy-four patients were assigned to Daikenchuto-treated group and one-hundred-sixty-two patients to placebo-treated group [134]. Mitsuo et al. reported that Daikenchuto decreased gastrointestinal dysmotility after hepatic resection in patients with liver cancer by accelerating the time to first bowel movement. In this phase III trial, two-hundred-nine patients were received 15 g/day, three times a day of Daikenchuto or placebo control from preoperative day three to postoperative day ten [135]. Kozo et al. also conducted a multicenter, randomized, double-blind, placebo-controlled trial to prove the efficacy of Daikenchuto. As a result, Daikenchuto promoted early recovery of postoperative bowel function during the immediate postoperative period after total gastrectomy. Data on one-hundred-ninety-five patients were finally included and evaluated in statistical analysis and patients received either Daikenchuto or matching placebo from postoperative days one to twelve [136]. Thirty-eight patients for Daikenchuto-treated group and thirty-three patients for placebo-treated group participated in a multicenter double-blind randomized placebo-controlled trial. Delayed gastric emptying is resolved by Daikenchuto but it has a limited effect on the resolution of postoperative paralytic ileus after open surgery in patients with sigmoid or rectosigmoid cancer [26]. Both green tea and black tea are derived from the leaves of the *Camellia sinensis* plant. To compare their efficacy, a prospective randomized, open label, three arm phase II intervention trial was carried out on ninety-three prostate cancer patients. As a result, green tea reduced nuclear NF-κB in radical prostatectomy tissue and systemic antioxidant activity but black tea did not compare to the water control [137]. Phase I/II study of gemcitabine-based chemotherapy plus curcumin was completed in japan by Kanai et al. Twenty-one gemcitabine-resistant patients with pancreatic cancer were treated with 8 g of curcumin daily and it was proved that curcumin sensitized the pancreatic cancer cells to gemcitabine [138]. According to the study of Marx et al., ginger effectively reduced the chemotherapy-induced nausea. A phase II randomized double blind placebo controlled trial showed that standardized ginger extract not only improved the chemotherapy-induced nausea but also reduced the cancer-related fatigue in patients undergoing moderately or highly emetogenic chemotherapies in Australia [139]. In a phase II randomized, double-blind clinical trial study which was completed in Iran, forty-five women with breast cancer undergoing chemotherapy participated and fifteen women were assigned to ginger-treated group, chamomile-treated group and untreated control group, respectively. This trial showed that frequency of nausea and vomiting-caused by breast cancer chemotherapy was reduced by capsules of powdered ginger root and *Matricaria chamomilla* extract [140]. Also, a phase II/III, randomized, double-blind, placebo-controlled clinical trial demonstrated that the severity of acute chemotherapy-induced nausea in adult cancer patients was significantly suppressed by ginger supplement at a daily dose of 0.5, 1.0 g per person. This trial included a total of five-hundred-seventy-six patients in final analysis and was completed [141]. It is reported that Hangeshashinto (TJ-14, a Kampo medicine) reduces the level of prostaglandin E2 and affects the cyclooxygenase activity. According to a double-blind, placebo-controlled, randomized phase II study which was completed in japan, Hangeshashinto alleviated chemotherapy-induced oral mucositis. Forty-three out of ninety eligible patients were assigned to Hangeshashinto-treated group and forty-seven assigned to placebo-treated group. Matsuda et al. proved that Hangeshashinto improved the mucositis-induced by chemotherapies in infusional fluorinated-pyrimidine-based colorectal cancer [142]. Aoyama et al. also reported that Hangeshashinto reduced the duration of any grade of oral mucositis-induced by gastric cancer chemotherapies. Ninety-one gastric cancer patients participated in this trial which was conducted in japan and it is currently no longer in recruiting status [143]. One of the most important limitations of cisplatin-based chemotherapies is nephrotoxicity which induces many complications such as ototoxicity, severe nausea and vomiting [144]. In a phase II/III double-blind, randomized clinical trial completed in Iran, one-hundred-twenty patients were randomly assigned to two groups, lycopene-treated group with standard regimen of kidney injury prevention and control group treated with only the standard regimen of kidney injury prevention. As a result, lycopene relieved the complications of cisplatin-induced nephrotoxicity by improving renal function [145]. A phase I/II randomized placebo-controlled double-blind clinical trial suggested that chemotherapy-induced oral mucositis was prevented by quercetin administration which has a biphasic anti-inflammatory effect. Twenty adult patients who underwent high dose chemotherapy for blood malignancies participated in this trial. Half of them were assigned to oral quercetin capsules-treated group and the others were assigned untreated control group [146]. Rikkunshito, an herbal medicine, treated the anorexia and functional dyspepsia and recovered the decreased food intake caused by cisplatin. Ohnishi et al. reported that it suppressed the chemotherapy-induced nausea, vomiting and anorexia induced by cisplatin and paclitaxel treatment. A randomized phase II study was carried out on thirty-six uterine cervical or corpus cancer patients and they were divided into rikkunshito-treated group and the untreated control group respectively [11]. *Aloe vera* has anti-inflammatory, bactericidal and wound healing effects. Several clinical trials were conducted to evaluate the efficacy of *Aloe vera* in treating radiation-induced complications. In a triple-blind, randomized, controlled clinical trial, twenty-six of head and neck cancer patients were divided into two groups: *Aloe vera* mouthwash as the intervention group and benzydamine mouthwash as the control group. It was proved that *Aloe vera* mouthwash was as beneficial as benzydamine mouthwash in decreasing the severity of radiation-induced mucositis [147]. Adeleh et al. also reported that *Aloe vera* ointment improved the symptoms of acute radiation-induced proctitis, including diarrhea and fecal urgency scores and improved the quality of life. Twenty patients who received pelvis radiotherapy completed in this trial. Eleven patients of them were allocated to the placebo ointment-treated group and nine of them to the *Aloe vera* ointment-treated group [148]. A randomized, double-blind, placebo-controlled clinical trial was carried out for radiotherapy-induced dermatitis on thirty breast cancer patients. The randomized patients took 2.0 g of curcumin or placebo orally three times a day throughout their course of radiotherapy. This trial showed that curcumin treatment, at 6.0 g daily, significantly alleviated the severity of radiation dermatitis and moist desquamation [149]. Xerostomia and oral mucositis are main side effects during and after the radiotherapy in head and neck cancer patients. Ghazaleh et al. proved that dry flowers of *Alcea digitata* Alef and *Malva sylvestris* improved the quality of life in head and neck cancer patients with radiation-induced xerostomia in a randomized, double-arm, open-label active-controlled clinical trial in Iran. Half of sixty patients were received sachets containing 4 g of mixed powder of *Alcea digitata* Alef and *Malva sylvestris* as the intervention group and Hypozalix spray (artificial saliva) as the control group respectively [150]. Melanie et al. proved the efficacy of thyme honey through a randomized controlled phase II trial. The data of this trial supported that thyme honey decreased the radiation-induced oral mucositis and improved the quality of life in head and neck cancer patients. Sixty-four patients randomly received either thyme honey oral rinses as the intervention group or normal saline oral rinses as the control group [151]. Cancer-related fatigue is one of the common symptoms of cachexia. Luke et al. conducted a randomized, double-blind, multicenter phase II trial to demonstrate the efficacy of Omega-3 polyunsaturated fatty acids in reducing cancer-related fatigue of breast cancer survivors. Contrary to the original hypothesis, Omega-6 polyunsaturated fatty acids supplementation significantly reduced proinflammatory markers in the TNF-α signaling pathway and cancer-related fatigue compared with Omega-3 polyunsaturated fatty acids supplementation in breast cancer patients [152].

## 7. Oxidative Stress and Cancer

Reactive oxygen species (ROS) are defined as oxygen-containing chemical species with reactive properties. These include the superoxide (O_2_^−^) and hydroxyl (HO∙) free radicals as well as non-radical molecules such hydrogen peroxide (H_2_O_2_). These molecules are principally derived from the oxygen that is consumed in various metabolic reactions occurring mainly in the mitochondria, peroxisomes and the endoplasmic reticulum (ER). Oxidative stress refers to an imbalance of the above oxidant species and antioxidant species such as CAT, GPx, SOD and so forth [153]. ROS play a dual role within tumor. On the one hand, ROS have the ability to promote tumorigenesis. ROS accumulation can promote cancer progression through activating transcriptional factors and regulating the expression of cell cycle genes. On the other hand, ROS also generate intracellular signals that stimulate cell death. Generation of excessive ROS promotes apoptosis, becoming a principle of anticancer targeted therapies such as chemotherapy and radiotherapy [154]. Thus, oxidative stress may have suppressive effect to tumor cells, while the additional oxidative stress induced by cancer treatments result in further complications. Several antineoplastic drugs currently used for cancer chemotherapy produce high levels of oxidative stress, which diminishes the efficacy of antineoplastic agents to kill cancer cells and causes damage to cellular components [15]. For example, taxanes, vinca alkaloids and antimetabolites stimulate release of cytochrome c from mitochondria and interfere the electron transport chain, resulting in cell death and O_2_^−^ production. Doxorubicin penetrates the inner membrane of cardiac mitochondria and induces O_2_^−^ release. 5-Fluorouracil (5-FU) generates mitochondrial ROS via a p53-dependent pathway. Other drugs also inhibit the ubiquitin–proteasome pathway and affect GSH, thioredoxin and glutamine metabolism. Ionizing radiation also leads to persistent oxidative stress. NADPH oxidase, another important source of ROS, is also activated by radiation exposure [153]. Radiation induces clustered DNA damage and double-strand break, contributing to cell death. In plasma membrane, radiation exposure produces lipid metabolites which involve in inflammatory response and cancer development. In mitochondria, excessive ROS generation by irradiation leads to permanent mitochondrial malfunction, resulting in cytochrome c release. Released cytochrome c in cytosol triggers the intrinsic apoptotic signaling by stimulating caspase 3/7 cascade activation. Upregulations of ER stress-associated genes such as protein kinase R (PKR)-like endoplasmic reticulum kinase (PERK), transcription factor 6 (ATF4), transcription factor 6 (ATF6), growth arrest and DNA damage 34 (GADD34) and inositol requirement 1 (IRE1) might contribute to adaptive survival signaling in cancer cells during radiotherapy [16]. Hight level of ROS is also an important factor in the development of muscle atrophy and fatigue, which are the main features of cachexia. In cancer cachexia patients, oxidative stress accelerates muscle wasting and cachexia progression. Oxidative stress modulates all the mechanisms involved in the development of skeletal muscle atrophy, such as ubiquitin proteasome system overactivation, protein synthesis pathway diminution, autophagy deregulation and increased myonuclear apoptosis [155]. In addition, ROS disturb the mitochondrial respiratory chain function within muscle, leading to a decreased ATP formation. This poor ATP level is a favorable condition for high mitochondrial ROS production, thereby maintaining the vicious circle. The mitochondrial energetic inefficiency and the subsequent accumulation of oxidative stress may impede the capacity of muscle to generate sufficient force and ensure basic physical needs. This ROS-dependent mechanism explains the increased fatigue observed in cachectic individuals [154]. Antioxidants are able to scavenge ROS and prevent harmful effects induced by oxidative stress. In this regard, antioxidant supplements can be used to prevent cancer and potentiate cancer therapy by providing protection against various sequela. Preventive and therapeutic effects of plant-derived antioxidants have been widely reported. 

## 8. Limitations of the Studies

There were some drawbacks in current studies. Over half of the reviewed studies about post-surgical complications were limited to gastrointestinal tract, liver and pancreas surgery. Also, studies about sequela of surgery were focused on clinical trials, while in vivo and in vitro mechanism studies accounted for a small proportion. Several shortcomings have been observed in current studies of radiotherapy complications. Information about cancer types, which can help to elaborate the detailed mechanisms of side effects, was not available in some studies. Short experiment duration in some studies can be another limitation. Kim et al. conducted the research injecting *Panax ginseng* water extract intraperitoneally for 4 days [106]. Study done by Rockwell et al. administered PHY906 for 4 days either [107]. Even though they were done in the mouse model, 4 days seems to be rather short period to demonstrate efficacy considering the toxicity of radiation-induced side effects. Also, clinical trials regarding radiotherapy sequela were slightly weighted on head and neck cancer patients. In cachexia, only 1 clinical trial was reviewed, although there were many in vivo and in vitro studies. More clinical trials are required to assess the efficacy and safety regarding cancer cachexia. Likewise, there was only 1 current study about fatigue and anemia, one of the symptoms of cancer cachexia. This indicates that current studies about cancer cachexia are still lacking. Some shortcomings in our research also exist. First, reviewed studies including extracts and herbal formulas account for a larger proportion than studies including single component. As extracts and herbal formulas may have different and more complex therapeutic mechanisms, unexpected variables are likely to exist, thus making it difficult to investigate the mechanism of individual component. Especially in studies reviewing cachexia, most researches deal with formulas and herb pairs. However, we should take into account that such multiple composition contributes to relieving complicated and multifaceted symptoms. Another point to consider is that the researches contrary to our reviewed studies were reported afterwards. In our reviewed studies, propolis and *Aloe vera* were reported to alleviate radiation-induced mucositis in Wistar rat model and phase II trial respectively [108,147]. However, Marucci et al. conducted double-blind randomized phase III study, reporting that some natural products including propolis and *Aloe vera* had no effect on acute mucositis during chemoradiotherapy for head and neck carcinoma [156]. Yet, many natural products such as curcumin, ginger, green tea, honey and Hangeshashinto and so forth still show consistent efficacy in both in vivo, in vitro and clinical trial. Moreover, some natural products were reported to impair the cancer pathogenicity. Spradlin et al. recently indicated that Nimbolide, a terpenoid natural product derived from the Neem tree, impairs breast cancer pathogenicity by disrupting RNF114 interactions, thus inhibiting p21 ubiquitination and degradation [157]. Further examination is needed to investigate whether any of the natural products reviewed in this study have potential side effects. Lastly, our research only included recent 10-years studies and English articles. This may weaken the comprehensiveness and diversity but rather strengthen the validity at the same time. 

## 9. Hypothesis of a Possible Trial 

Clinical trials for cancer treatment complications and cachexia are currently lacking compared to preclinical studies. Additional clinical trials are needed to develop effective treatments for cancer patients. Considering the aim of our study, possible trials can be developed for cancer patients starting from the preclinical evidences reviewed in this study. A clinical trial with ginger, extracted from *Zingiber officinale*, as a potential plant-derived drug can be considered. Ginger have shown consistent efficacy in different extraction methods (water extract, oleoresin, essential oil) and different experimental models (in vivo, in vitro) in preclinical studies. Ginger hydro-alcoholic extract suppressed oxidative DNA damage and inflammatory reactions after γ-ray exposure. Total antioxidant capacity was enhanced while 8-OhdG and CRP were downregulated in Wistar rat model when pretreated for 10 days at a dose of 50 mg/kg [101]. Ginger oleoresin mitigated radiation-induced cell injury in human mesenchymal stem cells at a dose of 1, 10, 100, 1000 μg/mL for 24, 48, 72 h. Increased levels of HO-1 and NQO-1 with decreased levels of ROS were observed [102]. Ginger essential oil administration on Balb/C mice at a dose of 100 and 500 mg/kg inhibited γ-irradiation induced oxidative stress. Increased levels of SOD, catalase, GPx and GSH indicated its antioxidative property [103]. Considering that ginger has reported considerable efficacies in various extraction methods and experimental models, we may hypothesize a clinical trial to confirm the antioxidative, anti-inflammatory, cytoprotective effect of ginger for cancer patients. Regarding the clinical trial design, two-arm, parallel double-blinded randomized controlled trial is preferred. This may be helpful to examine the primary endpoint of this trial, which is to determine whether the cancer therapy sequela and cachexia are effectively suppressed in the treatment group compared to the placebo group.

## 10. Conclusions

In conclusion, our research reviewed the efficacy of plant-derived antioxidants in sequela of cancer-related treatments and cachexia. We focused on resolving the side effects of existing treatments and cachectic symptoms in relation to oxidative stress and inflammation, rather than direct anti-cancer effect. The significance of our review is that it is a pioneering study covering both cancer treatment sequela and cancer cachexia. While various anti-cancer drugs and frontline cancer treatments have already been developed sufficiently, researches in minimizing and managing their side effects still lack attention. Especially, cachexia is yet underestimated and related studies are even more insufficient. This study will serve as a valid and comprehensive database regarding plant-derived antioxidants as well as an opportunity to trigger the interest in cancer treatment complications and cancer cachexia. 

## Figures and Tables

**Figure 1 antioxidants-09-00836-f001:**
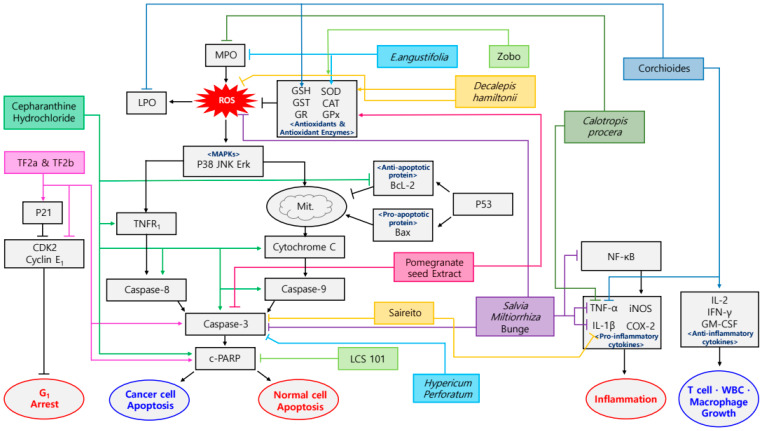
Schematic diagram of anti-inflammatory & antioxidant mechanisms of Plant-derived drugs in sequela of chemotherapy. TF2a, theaflavin-3-gallate; TF2b, theaflavin-3′-gallate.

**Figure 2 antioxidants-09-00836-f002:**
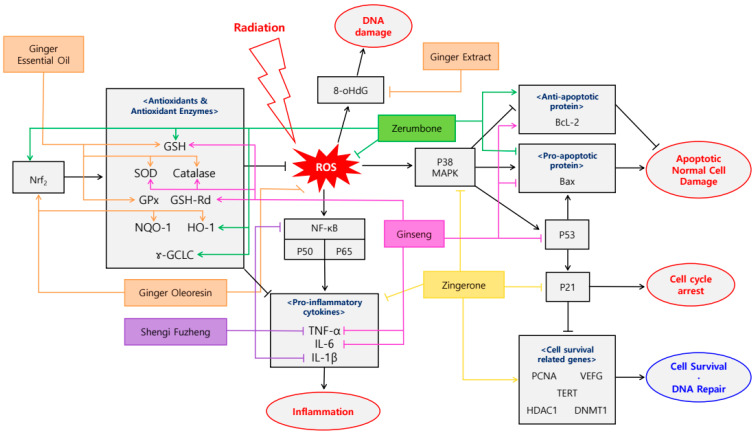
Schematic diagram of anti-inflammatory & antioxidant mechanisms of plant-derived drugs in sequela of radiotherapy.

**Figure 3 antioxidants-09-00836-f003:**
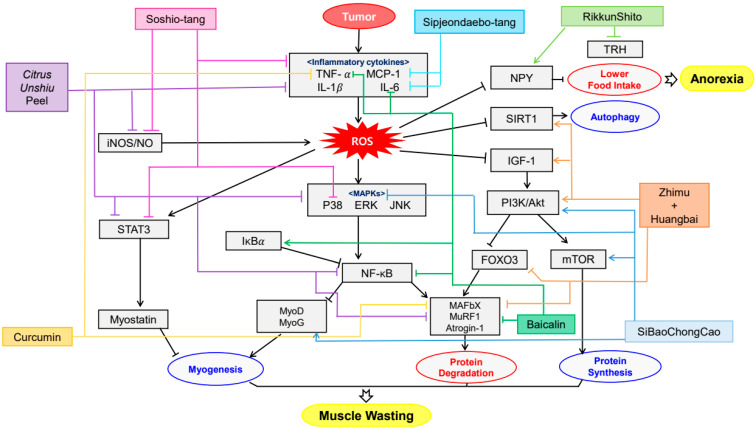
Schematic diagram of anti-inflammatory & antioxidant mechanisms of plant-derived drugs in cachexia.

**Table 1 antioxidants-09-00836-t001:** Plant-derived drugs and sequela of surgery.

Surgery	Compound/Extract	Source	Cancer type	Cell Line/Animal Model	Dose; Duration	Efficacy	Mechanism	Reference
Colorectal polyp removal surgery	Epigallocatechin gallate green tea extract	*Amellia sinensis* O. Kuntze	Colon adenoma	Human	150 mg; 3 years	Prevention of metachronous adenomas in colorectum	↓P450 isoenzymes	[23]
Hepatocellular tumor xenograft surgery	2′,4′-Dihydroxy-6′-methoxy-3′,5′-dimethylchalcone	*Cleistocalyx operculatus* (Roxb.)	Hepatocellular tumor cancer	BEL-7402/5-FU, nude mice	40 mg/kg; 16 days	Decrease of 5-FU resistance	↑caspase-3	[24]
Pancreatic cancer xenografts surgery	Nemorosone		Pancreatic cancer	NMRI nu/nu mice	50 mg/kg; 1 day	Decrease of side-effects and cancer growth	↑CYP3A4	[25]
Sigmoid or rectosigmoid cancer surgery	Daickenchuto		Sigmoid or rectosigmoid cancer	Human	15.0 g; 6 days	Decrease of delayed gastric emptying and paralytic ileus		[26]
Sigmoid colon cancer surgery	Daickenchuto		Sigmoid colon cancer	Human	7.5 g; 2 h	Increase of postoperative intestinal motility	↑CRP	[27]

EGCG, epigallocatechin gallate; DMC, 2′,4′-Dihydroxy-6′-methoxy-3′,5′-dimethylchalcone; 5-FU, 5-fluorouracil; CYP3A4, Cytochrome P450 3A4; CRP, C-Reactive protein; ↑, up-regulation; ↓, down-regulation.

**Table 2 antioxidants-09-00836-t002:** Plant-derived drugs and sequela of chemotherapy.

Chemotherapy	Compound/Extract	Source	Cancer Type	Cell Line/Animal Model	Dose; Duration	Efficacy	Mechanism	Reference
Carboplatin	Curcumin	*Curcuma longa* Linn		T241-bearing mice	5 mM; 15 days	Alleviation of myelosuppression	↑BRCA1, BRCA2, ERCC1	[62]
Cisplatin	*Bacopa monnieri* extract	*Bacopa monnieri*		Pigeon	10, 20, 40 mg/kg; 24 h	Alleviation of vomiting	↓dopamine, 5-HT, 5-HIAA	[63]
Cisplatin	Cepharanthine hydrochloride		Esophageal squamous cell carcinoma	Eca109 cells inoculated BALB/c nude mice	10 mg/kg; 13 days	Increase of anticancer properties and decrease of side effects	↑c-PARP, c-caspase-3, 8, 9, TNFR1, p-JNK, cytochrome C ↓Bcl-2	[64]
Cisplatin	Pomegranate seed extract	*Punica granatum*		SD rats	300 mg/kg; 15 days	Alleviation of acute nephrotoxicity and hepatotoxicity	↑GSH, GST, GPx, SOD ↓Lipid peroxidation, MDA, caspase-3	[65]
Cisplatin	Theaflavin-3-gallate, theaflavin-3′-gallate		Ovarian cancer	A2780/CP70, IOSE-364	5, 10, 20 µM; 24 h	Prevention of ovarian cancer	↑caspase-3, -7, c-PARP, p21, p38, p53 ↓CDK2, CDK4, cyclin E1	[66]
Cyclophosphamide	Avocado methanol extract	*Persea Americana* Mill		Human lymphocyte	100, 200, 300 mg/kg; 70 h	Decrease of chromosomal aberrations		[67]
Cyclophosphamide	*Curculigo orchioides* methanolic extract	*Curculigo orchioides* Gaertn.		Balb/c, Swiss albino mice	25, 50, 100, 200 mg/kg; 14 days	Increase of anticancer properties and decrease of side effects	↑α-esterase, IL-2, GM-CSF, IFN-γ, GSH ↓TNF-α, LPO, ALP, GPT	[68]
Cyclophosphamide	*Decalepis hamiltonii* methanolic extract	*Decalepis hamiltonii*		BALB/c mice	0.5 mg; 10 days	Alleviation of side effects	↑GSH ↓SGOT, SGPT	[69]
Cyclophosphamide	*Decalepis hamiltonii* aqueous extract	*Decalepis hamiltonii* Wight & Arn		Swiss albino mice	50, 100 mg/kg; 10 days	Alleviation of side effects	↑GSH, SOD, CAT, GPx, GR, GST ↓ROS	[70]
Docetaxel	(1) Green tea (2) Quercetin	*Camellia sinensis*	Prostate cancer	LAPC-4-AI, PC-3	(1) 40, 5 μM; 48 h	Increase of therapeutic effect and decrease of chemoresistance	↑Bax/Bcl-2 ↓PI3K/Akt, STAT3, MRP1, CD44+/CD24−	[71]
Doxorubicin	*Byrsonima verbascifolia* water extract	*Byrsonima verbascifolia*		Somatic cells of Drosophila melanogaster	25, 50, 100 mg/mL; 2 days	Alleviation of doxorubicin-induced damage		[72]
Doxorubicin	Ginsenoside Rh2		Breast cancer	MDA-MB-231	20 μg/mL; 48 h	Alleviation of cellular senescence	↓Vimentin, beta-catenin, Snail, caspase 3/7, MCP-1, CXCL1, IL-6, IL-8, p-MEK1, p-p38, p-STAT3, p-NF-κB p65	[73]
				MCF-10A			↓caspase 3/7, MCP-1, CXCL1, IL-6, IL-8, p-p38, p-STAT3	
Doxorubicin, 5-fluorouracil	LCS101	*Astragalus membranaceus, Atractylodes macrocephala, Citrus reticulate, Glehnia littoralis, Ligustrum lucidum, Lycium chinense, Milletia reticulata, Oldenlandia diffusa, Ophiopogon japonicus, Paeonia lactiflora, Paeonia obovata, Poriae cocos, Prunella vulgaris* *, Scutellaria barbata*	Breast, Colorectal, Prostate cancer	MCF7, MDA-MB-231, HCT116, PC-3, DU-145, MCF10A, EP#2	1, 2, 3 mg/mL; 24, 48, 72 h	Regulation of tumorigenic and non-tumorigenic cells	↓c-PARP-1	[74]
5-Fluorouracil	*Calotropis procera* latex	*Calotropis procera*		Golden hamsters	0.25, 1, 5, 25 mg/kg; 24 h before, 24 h after mechanical trauma	Alleviation of oral mucositis	↓COX-2, iNOS, TNF- α, IL-1β, MPO	[75]
5-Fluorouracil	Ciji Hua’ai Baosheng Granule Formula	*Radix Codonopsis, Radix astragali Mongolici, Bulbus fritillariae Thunbergii, Rhizoma Arisaematis Erubescentis, Pericarpium Citri Reticulatae, Poria, Cortex Magnoliae Officinalis, Fructus Aurantii Submaturus, Rhizoma Atractylodis Macrocephalae, Fructus Amomi, Fructus Alpiniae Oxyphyllae, Semen Lablab Album, Fructus Hordei Germinatus, Rhizoma sparganii, Spina Gleditsiae, Cortex Albiziae, Concha Ostreae, Ganoderma Lucidum, Fructus Psoraleae*	Hepatic cancer	Kunming mice	16, 32, 64 g/kg; 21 days	Alleviation of tumor growth and appearance		[76]
5-Fluorouracil	*E. angustifolia* hydroalcoholic extract	*Echinacea angustifolia* de Candolle		Golden hamsters	3000 mg/kg; 5 days	Healing stimulatory and anti-inflammatory properties in oral mucositis	↑SOD ↓MPO	[77]
5-Fluorouracil	Saireito			C57BL/6 mice	100, 300, 1000 mg/kg; 6 days	Prevention of intestinal mucositis	↓Caspase-3, TNF-α, IL-1β	[78]
5-Fluorouracil	*Salvia miltiorrhiza* Bunge	*Salvia miltiorrhiza* Bunge		Golden Syrian hamsters	100, 500, 1000 mg/kg; 10 days	Prevention of oral mucositis	↑DPPH ↓NF-κB, caspase-3, ROS, TNF-α, IL-1β	[79]
Methotrexate	*Hibiscus sabdariffa* extract (Zobo)	*Hibiscus sabdariffa* Linn		Albino Wistar rats	10 mL/kg; 14 days	Alleviation of oxidative hepatotoxicity	↑SOD, CAT, GPx ↓MDA	[80]
Oxaliplatin	Toad skin aqueous extract (Huachansu)	*Bufo bufo gargarizans* Cantor		SD rats	1.25, 2.5 g/kg; 21 days	Prevention of allodynia and hyperalgesia	↑TRPV4 ↓TRPV1	[81]
Oxaliplatin	*Hypericum perforatum* hydrophilic extract	*Hypericum perforatum* Linn		Rat astrocytes, HT-29	5, 50, 250 μg/mL; 4, 8 h	Alleviation of chemotherapy-induced neuropathy	↓Caspase-3	[82]
Taxane	Annurca Apple polyphenolic Extract	*Malus Pumila* Miller cv. Annurca		Wild-type C57BL/6 mice	400 mg/L; 7 days	Protection of murine hair follicles from dystrophy	↑PGF2α ↓PPP	[83]
TS-1	Juzentaihoto	*Astragali Radix, Cinamomi Cortex, Rehmanniae Radix, Paeoniae Radix, Cnidii Rhizoma, Angelicae Radix, Ginseng Radix, Hoelen, Glycyrrhizae Radix, Atractylodis Lanceae Rhizoma*		Balb/c mice	1 g/kg; 3, 5, 7 days	Activation of hematopoiesis	↑CD34+ BMC ratio ↓WBC	[59]
Vincristine	*Ginkgo biloba* extract	*Ginkgo biloba*		SD rats	50, 100, 150 mg/kg; 15, 30, 60, 90, 120, 150, 180 m	Alleviation of mechanical and cold hyperalgesia		[84]

BRCA, breast cancer gene; ERCC, excision repair cross-complementing; 5-HT, 5-hydroxytryptamine; 5-HIAA, 5-hydroxyindoleacetic acid; PARP, poly ADP ribose polymerase; TNFR1, tumor necrosis factor receptor 1; JNK, Jun N-terminal kinase; Bcl-2, B-cell lymphoma 2; MDA, malondialdehyde; CDK, cyclin-dependent kinase; IL, interleukin; GM-CSF, granulocyte monocyte colony-stimulating factor; IFN, interferon; GSH, glutathione; TNF-α, tumor necrosis factor-α; LPO, lipid peroxidation; ALP, alkaline phosphatase; GPT, glutamate pyruvate transaminase; SGOT, serum glutamate oxaloacetate transaminase; SGPT, serum glutamate pyruvate transaminase; SOD, superoxide dismutase; CAT, catalase; GPx, glutathione peroxidase; GR, glutathione reductase; GST, glutathione-S-transferase; ROS, reactive oxygen species; Bax, Bcl-2 associated x protein; PI3K, phosphoinositide 3-kinase; Akt, protein kinase B; STAT3, signal transducer and activator of transcription 3; MRP1, multidrug resistance-related protein 1; ST, standard; HB, high bioactivation; MCP-1, monocyte chemoattractant protein 1; CXCL1, chemokine (C-X-C motif) ligand 1; MEK1, mitogen-activated protein kinase kinase 1; NF-κB, nuclear factor kappa-light-chain-enhancer of activated B cells; COX-2, cyclooxygenase-2; iNOS, inducible nitric oxide synthase; MPO, myeloperoxidase; DPPH, 2-diphenyl-1-picrylhydrazyl; TRPV, transient receptor potential vanilloid; PGF2α, prostaglandin F 2α; PPP, protein phosphatase; BMC, bone marrow cell; ↑, up-regulation; ↓, down-regulation.

**Table 3 antioxidants-09-00836-t003:** Plant-derived drugs and sequela of radiotherapy.

Compound/Extract	Source	Cancer Type	Target Tissue	Cell Line/Animal Model	Dose; Duration	Efficacy	Mechanism	Reference
Bojungikki-tang water extract	*Panax ginseng* C. A. Meyer, *Atractylodes macrocephala* Koidz., *Astragalus membranaceus* Bunge, *Angelicae gigantis* Radix, *Citrus aurantium* Linne, *Ziziphus jujuba* var. inermis, *Bupleurum falcatum* Linne, *Glycyrrhiza uralensis* Fisch., *Zingiber officinale* Roscoe, *Cimicifuga heracleifolia* KOM.			Human	9 g; 4 weeks	Restoration of B cells decrease		[98]
Cranberry capsules	*Vaccinium macrocarpon* Ait.	Prostate cancer	Urothelium	Human	72 mg/day; 2 weeks	Alleviation of cystitis		[99]
Cranberry	*Vaccinium macrocarpon* Ait.	Prostate cancer	Prostate	Human	200 mg; 6, 7 weeks	Alleviation of lower urinary tract infections and urinary symptoms		[100]
Ginger hydro-alcoholic extract	*Zingiber officinale*		Kidney	Wistar rats	50 mg/kg; 10 days	Prevention of γ-ray induced kidney damage	↑TAC ↓8-OhdG, CRP	[101]
Ginger oleoresin	*Zingiber* *officinale*		Human mesenchymal stem cells	Human	1, 10, 100, 1000 μg/mL; 24, 48, 72 h	Prevention of cell injury	↑HO-1, NQO-1 ↓ROS	[102]
Ginger essential oil	*tabke*			Balb/C mice	500 mg/kg; 19 days	Prevention of γ-irradiation induced damage	↑SOD, catalase, GPx, GSH, MNPCE, MNNCE	[103]
Hangeshashinto (TJ-14)	*Pinellia ternata, Scutellaria baicalensis, Zingiber officinale* Roscoe, *Glycyrrhiza uralensis* Fisch., *Zizyphus jujuba* Mill., *Panax ginseng* C. A. Meyer, *Coptis chinensis* Franch.	Head, neck cancer	Buccal mucosa	Syrian golden hamsters	2%; 28 days	Prevention of radiation mucositis	↓COX-2	[104]
Pure natural honey (Dabur honey)	*Apis mellifera* Linnaeus	Head, neck cancer	Head, neck	Human	60 m/L; 7 weeks	Alleviation of radiation mucositis		[105]
*Panax ginseng* water extract	*Panax ginseng* C. A. Meyer		Liver	C57BL/6N mice	25, 50, 100 mg/kg; 4 days	Prevention of liver injury	↑TAC, GSH, GSH-Rd, SOD, catalase, Bcl-2, Bcl-xL ↓4-HNE, ROS, MDA, TNF-α, IL-6, p53, Bax	[106]
PHY906	*Scutellaria baicalensis*, *Glycyrrhiza uralensis* Fisch., *Paeonia lactiflora, Ziziphus jujub*a var. inermis	N/A	Abdomen	EMT6, BALB/c Rw mice	500 mg/kg; 4 days	Alleviation of abdominal irradiation induced toxicity		[107]
Propolis ethanolic extract	*Apis mellifera* Linnaeus		Head, neck	Wistar rats	100, 200 mg/kg; 10 days	Alleviation and delay of mucositis		[108]
Shenqi Fuzheng	*Codonopsis pilosula, Astragalus membranaceus* Bunge		Brain	C57BL/6J mice	20 mL/kg/d; 28 days	Alleviation of brain injury	↓HRP, TNF-α, IL-1β, NF-κB, PIDD-C, PIDD-CC, p65	[109]
Zerumbone	*Zingiber zerumbet* Smith		Skin	HaCaT	2–10 μM; 24 h	Dermato-protective efficacies	↑Bcl-2, Nrf2, HO-1, γ-GCLC, GSH, p38 MAPK, PI3K ↓LDH, ROS, Bax	[110]
				BALB/c-nu mice	55, 110 μg/day; 14 days			
Zhuye Shigao Granule	*Lophatherum gracile* Brongn., *Gypsum, Panax ginseng* C. A. Meyer, *Liriope platyphylla, Pinellia ternate* (Thunb.) Breit., *Glycyrrhiza uralensis* Fisch., *Rabdosia serra* (Maxim.) Hara, *Hedyotis diffusa* Willd., *Scutellaria barbata* D. Don, *Coix lacryma-jobi*, *Curcuma longa* Linne	Lung, esophagus, mediastinal cancer	Chest, mediastinum	Human	12 mg; 4 weeks	Alleviation of acute esophagitis		[111]
Zingerone	*Zingiber* *officinale*			Albino rats	25 mg/kg; 21 days	Prevention of cardiotoxicity	↑GSH, CAT, ETC complex I, II, IV ↓cTnT, LDH, CK-MB, MDA, TNF-α, MPO, caspase-3	[29]
Zingerone	*Zingiber* *officinale*		Skin	Keratinocyte stem cells	10, 20, 100 μM; 24 h	Prevention of UVB-induced keratinocyte damages	↑PCNA, VEGF, TERT, HDAC1, DNMT1 ↓TNF-α, IL-1β, IL-6, p21, p42/44 MAPK, p38 MAPK	[112]
Gel containing *Pinus halepensis* bark aqueous extract	*Pinus halepensis* Mill.	Breast cancer	Skin	SKH-HR2 hairless mice	5%; 60 days	Alleviation of chronic and granulomatous inflammation		[113]
Ointment containing marine isopod *Ceratothoa oestroides* olive oil extract	*Ceratothoa oestroides* Risso.	Breast cancer	Skin	SKH-HR2 hairless mice	10%; 60 days	Alleviation of chronic and granulomatous inflammation		[113]

TAC, total antioxidant capacity; 8-OhdG, 8-hydroxy-2′-deoxyguanosine; CRP, C-Reactive protein; hMSCs, human mesenchymal stem cells; HO-1, heme oxygenase-1; NQO-1, NAD(P)H:quinone oxidoreductase 1; ROS, reactive oxygen species; SOD, superoxide dismutase; GPx, glutathione peroxidase; GSH, glutathione; MNPCE, micronucleated polychromatic erythrocytes; MNNCE, micro nucleated normochromatic erythrocytes; P/N ratio, polychromatic erythrocytes/normochromatic erythrocytes ratio; COX-2, cyclooxygenase-2; WBC, white blood cells; GSH-Rd, glutathione reductase; Bcl-2, B-cell lymphoma 2; Bcl-xL, B-cell lymphoma-extra-large; 4-HNE, 4-hydroxynonenal; MDA, methylenedioxyamphetamine; ALT, alaninetransaminase; ALP, alkalinephosphatase; TNF-α, tumor necrosis factor-α; IL-6, interleukin-6; p53, tumor protein 53; Bax, Bcl-2-associated X protein; HRP, horseradish peroxidase; IL-1β, interleukin-1β; NF-κB, nuclear factor kappa-light-chain-enhancer of activated B cells; PIDD-C, C-terminal fragment of p53-induced protein segments with a death domain; PIDD-CC, the twice-cleaved fragment of p53-induced protein with a death domain; Nrf2, nuclear factor-E2-related factor-2; γ-GCLC, γ-glutamyl cysteine ligase; MAPK, mitogen activated protein kinase; PI3K, phosphoinositide 3-kinases; LDH, lactate dehydrogenase; CAT, catalase; ETC, electron transport chain; cTnT, cardiac troponin T; CK-MB, creatine kinase-MB; MDA, malondialdehyde; MPO, myeloperoxidase; PCNA, proliferating cell nuclear antigen; VEGF, vascular endothelial growth factor; TERT, telomerase reverse transcriptase; HDAC1, histone deacetylase 1; DNMT1, DNA (cytosine-5)-methyltransferase 1; ↑, up-regulation; ↓, down-regulation.

**Table 4 antioxidants-09-00836-t004:** Plant-derived drugs and cachexia.

Compound/Extract	Source	Cell Line/Animal Model	Dose; Duration	Efficacy	Mechanism	Reference
Baicalin	*Scutellaria baicalensis*	CT26 adenocarcinoma inoculated BALB/c mice	50, 150 mg/kg; 15 days	Amelioration of anorexia, weight loss and muscle atrophy	↑IκBα ↓NF-κB, TNF-α, IL-6, MURF1, Atrogin-1, p65	[126]
*Citrus unshiu* peel water extract	*Citrus unshiu* Markovich	CT26 adenocarcinoma-induced cancer cachexia BALB/c mice	250, 500 mg/kg; 17 days	Amelioration of weight loss, muscle wasting and Hb loss	↑MyH, p-Akt ↓MAFbx, MuRF-1, IL-6, NO, iNOS, IL-1β, TNF-α, p-p38, ERK, JNK, IκBα, STAT3, p-p65,	[127]
Curcumin green tea extract	*Curcuma longa,* *Camellia sinensis*	C2C12 myotubes	10 μg/mL; 24 h	Amelioration of weight loss and muscle wasting	↓20S proteasome subunits, p42, MuRF1, MAFbx, PIF, TNF-α	[128]
Rikkunshito	*Atractylodes lancea, Panax ginseng, Pinellia ternate, Poria cocos, Zizyphus jujuba, Citrus unshiu, Glycyrrhiza uralensis, Zingiber officinale*	85As2 cells inoculated F344/NJcl-rnu/rnu rats	1 g/kg/day; 7 days	Amelioration of anorexia and weight loss	↑NPY ↓TRH	[129]
SiBaoChongCao	Cordyceps sinensis	C26 tumor-bearing BALB/c mice	1, 2 g/kg; 20 days	Amelioration of weight loss, muscle wasting and adipocyte cell reduction	↑MHC, MyoD, MyoG, p-AKT, p-mTOR, AMPKα, ERK, TORC1, PGC-1α ↓IL-6, TG, AMPK, p38 MAPK, p-HSL, UCP1	[130]
Sipjeondaebo-tang	*Angelica gigas, Astragalus membranaceus, Atractylodes japonica, Cinnamomum cassia, Cnidium officinale, Glycyrrhiza uralensis, Paeonia lactiflora, Panax ginseng, Poria cocos, Rehmannia glutinosa*	CT-26 inoculated- BALB/c mice	6.784, 67.84, 678.4 mg/kg; 21 days	Amelioration of anorexia, weight loss, muscle wasting and anemia	↓IL-6, MCP-1, PYY, GLP-1	[131]
Soshio-tang	*Bupleurum falcatum, Glycyrrhiza uralensis, Panax ginseng, Pinellia ternata, Scutellaria baicalensis, Zingiber officinale, Ziziphus jujuba*	J774A.1 macrophage cell line inoculated CT-26-bearing mice	50, 100 mg/kg; 18 days	Alleviation of weight loss, muscle wasting and appetite loss	↓NO, iNOS, IL-6, IL-1α, IL-1β, TNF-α, p38, NF-κB, IκBα, IKKαβ, STAT3	[132]
Zhimu and Huangbai herb pair	*Anemarrhena asphodeloides, Phellodendron amurense*	colon-26 adenocarcinoma inoculated C57BL/6 mice	104 mg/kg; 18 days	Amelioration of weight loss and muscle wasting	↑IGF-1, Akt, LC3B, SIRT1 ↓TNF-α, IL-6, atrogin-1, MuRF1, FOXO3	[133]

IκBα, inhibitor of kappa B; NF-κB, nuclear factor kappa B; TNF-α, tumor necrosis factor- α; IL-6, interleukin-6; MURF-1, muscle RING-finger protein-1; Hb, hemoglobin; MyH, myosin heavy chain; p-Akt, phospho-protein kinase B; MAFbx, muscle atrophy F-box; NO, nitric oxide; iNOS, inducible nitric oxide synthase; ERK, extracellular signal-regulated kinase; JNK, c-Jun N-terminal kinase; STAT3, signal transducer and activator of transcription 3; PIF, proteolysis-inducing factor; NPY, Neuropeptide Y; TRH, thyrotropin-releasing hormone; PVN, paraventricular nucleus; MHC, myosin heavy chain; MyoD, myogenic differentiation antigen; MyoG, myogenic regulatory factors; mTOR, mammalian target of rapamycin; AMPK, adenosine monophosphate-activated protein kinase; TORC1, target of rapamycin kinase complex1; PGC-1α, peroxisome proliferator-activated receptor gamma coactivator 1α; TG, triglyceride; MAPK, mitogen-activated protein kinase; HSL, hormone-sensitive lipase; UCP1, uncoupling protein 1; MCP-1, monocyte chemoattractant protein-1; PYY, peptide YY; GLP-1, glucagon like peptide-1; IGF-1, insulin-like growth factor 1; LC3B, microtubule-associated protein 1A/1B-light chain 3; SIRT1, sirtuin1; FOXO3, forkhead box O-3; ↑, up-regulation; ↓, down-regulation.

**Table 5 antioxidants-09-00836-t005:** Clinical trials.

Treatment	Compound/Extract	Source	Phase	Patients	Status	Number	Efficacy	Reference
Surgery	Daikenchuto	*Zingiber officinal*e Roscoe, *Zanthoxylum piperitum* De Candolle, *Panax ginseng* Carl Anton Meyer, maltose	III	336	Completed	UMIN000001592	Improvement of gastrointestinal dysfunction	[134]
Surgery	Daikenchuto	*Zingiber officinal*e Roscoe, *Zanthoxylum piperitum* De Candolle, *Panax ginseng* Carl Anton Meyer, maltose	III	209	Completed	UMIN000003103	Improvement of gastrointestinal dysmotility	[135]
Surgery	Daikenchuto	*Zingiber officinal*e Roscoe, *Zanthoxylum piperitum* De Candolle, *Panax ginseng* Carl Anton Meyer, maltose	III	195	Completed	UMIN000004693	Improvement of postoperative bowel function	[136]
Surgery	Daikenchuto	*Zingiber officinal*e Roscoe, *Zanthoxylum piperitum* De Candolle, *Panax ginseng* Carl Anton Meyer, maltose	III	71	Completed	UMIN000001793	Alleviation of postoperative paralytic ileus	[26]
Surgery	Green tea	*C*amellia sinensis	II	93	Active, not recruiting	NCT00685516	Increase of systemic antioxidant activity	[137]
Chemotherapy	Curcumin	*Curcuma longa* Linn	I, II	21	Completed	UMIN000001386	Sensitization of pancreatic cancer cells to gemcitabine	[138]
Chemotherapy	Ginger	*Zingiber officinale* Roscoe	II	34	Completed	ACTRN12613000120774	Improvement of chemotherapy-induced nausea	[139]
Chemotherapy	Ginger, Matricaria Chamomilla extract	*Zingiber officinale* Roscoe, *Matricaria chamomilla* Linné	II	45	Completed	IRCT2013020912404N1	Alleviation of nausea and vomiting	[140]
Chemotherapy	Ginger purified liquid extract	*Zingiber officinale* Roscoe	II, III	576	Completed	NCT00040742	Alleviation of acute nausea	[141]
Chemotherapy	Hangeshashinto	*Pinellia ternata* Breitenbach, *Scutellaria baicalensis* Georgi, *Glycyrrhiza uralensis* Fischer, *Zizyphus jujuba* Miller var. *inermis* Rehder, *Panax ginseng* Carl Anton Meyer, *Zingiber officinale* Roscoe, Coptis rhizome	II	90	Completed	UMIN000004287	Improvement of mucositis	[142]
Chemotherapy	Hangeshashinto	*Pinellia ternata* Breitenbach, *Scutellaria baicalensis* Georgi, *Glycyrrhiza uralensis* Fischer, *Zizyphus jujuba* Miller var. *inermis* Rehder, *Panax ginseng* Carl Anton Meyer, *Zingiber officinale* Roscoe, Coptis rhizome	II	91	No longer recruiting	UMIN000004214	Alleviation of oral mucositis	[143]
Chemotherapy	Lycopene	*Momordica cochinchinensis* Spreng, *Elaeagnus umbellata*, *Lycopersicon esculentum* etc.	II- III	120	Completed	IRCT2016050427745N1	Alleviation of nephrotoxicity related complications	[145]
Chemotherapy	Oral quercetin capsules	Plant flavonol from the flavonoid group of polyphenols	I, II	20	Completed	NCT01732393	Prevention of oral mucositis	[146]
Chemotherapy	Rikkunshito	*Atractylodes lancea* De Candlle, *Panax ginseng* Carl Anton Meyer, *Pinellia ternata* Breitenbach, *Poria cocos* Wolf, *Zizyphus jujuba* Miller var. *inermis* Rehder, *Citrus reticulata* Blanco, *Glycyrrhiza uralensis* Fischer, *Zingiber officinale* Roscoe	II	36	Completed	UMIN000011227	Improvement of nausea, vomiting and anorexia	[11]
Radiotherapy	*Aloe vera*	*Aloe ver*a	II	26	Completed	IRCT2012072410377N1	Alleviation of mucositis	[147]
Radiotherapy	*Aloe vera* ointment	*Aloe vera*	II	20	Completed	IRCT201606042027N6	Improvement of acute proctitis	[148]
Radiotherapy	Curcumin C3 Complex	*Curcuma longa* Linn	II	30	Completed	NCT01042938	Alleviation of dermatitis	[149]
Radiotherapy	Dry flowers of *Alcea digitata* Alef, *Malva sylvestris*	*Alcea digitata* Alef, *Malva sylvestris* Carl Linnaeus	II	60	Completed	NCT02854358	Improvement of xerostomia	[150]
Radiotherapy	Thyme honey	*Thymus Capitatus*, *Thymus Vularis*, *Thymus Serpyllum* etc.	II	64	Completed	NCT01465308	Improvement of oral mucositis	[151]
Cachexia	Omega-6 polyunsaturated fatty acids	Soybean oil	II	81	Completed	NCT02352779	Alleviation of cancer-related fatigue	[152]

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
