# Peer review of "Plant Extracts as Possible Agents for Sequela of Cancer Therapies and Cachexia"

_antioxidants, 2020, doi:10.3390/antiox9090836_

Round 1
Reviewer 1 Report
The review presented by Lee et al. represents a good collection of data concerning the role of plant extracts and herbal formulas on the sequela of cancer therapies. The presence of tables and figures is helpful in summarizing the data reported and in clarifying the mechanism of action of the molecules. However, to improve the readability a strong rearrangement must be done.
- Despite in the abstract there is this sentence “The primarymechanism of cancer sequela and cachexia is closely related to reactive oxygen species (ROS) and inflammation”, a very short chapter is present about Oxidative stress and cancer. This aspect must be examined more in depth.
- To avoid repeat of the same concepts results and discussion sections must be combined. In this way, the side effects of the therapies (the context) and the possible role of the plant extracts in that context can be discussed and understood. A further classification of the side effects of the therapies and the corresponding herbal formula or plant extract can improve the readability, otherwise it sounds simply like a list. I mean after discussing the cancer therapy (for example chemotherapy) a paragraph for the side effects (emesis, hepatotoxicity, skeletal muscle dysfunction, etc) and their corresponding “therapeutic support” should be added.
- The tables are useful, but the formatting must be ameliorated.
- A reference should be added: after lines 111-114; after lines 121-127.
- Bibliography must be checked: journal abbreviations are often incorrect; the pages are often lacking. Check reference number: 2, 17-19, 24-25, 28-31, 33, 42-43, 45, 54, 64, 75, 89, 91, 96-105, 108-118, 123, 129-132, 138.
Extensive editing of English language and style is required.
Just as an example: after the period do not start with And (see lines 169, 180, 428); damage with DNA at line 99; at line 83 change use with uses and at line 81 add in It is usually used addition;
Typos: Line 138 caner is cancer; lines 196 and 197 Cepharanthine hydrochloride
Add a space at lines 173 creative protein; Withaniasomnifera at line 179;
At line 240 substitute D.M. with Surname..
At line 256 Ana Paula is the name, use the Surname
At line 332 was treated to the patients in incorrect: the patients were treated.
At line 379 the following treatment.
At line 577 remove were before randomly received
At lines 578-579 “Cancer-related fatigue is one of the common cachexia. symptoms?
Author Response
We appreciate editors and reviewers for critical comments to improve the quality of our manuscript (antioxidants-913920), titled “Plant extracts as possible agents for sequela of cancer therapies and cachexia”. We earnestly responded to the raised comments point by points.
The review presented by Lee et al. represents a good collection of data concerning the role of plant extracts and herbal formulas on the sequela of cancer therapies. The presence of tables and figures is helpful in summarizing the data reported and in clarifying the mechanism of action of the molecules. However, to improve the readability a strong rearrangement must be done.
(Response): Thank you for careful comments.
Despite in the abstract there is this sentence “The primarymechanism of cancer sequela and cachexia is closely related to reactive oxygen species (ROS) and inflammation”, a very short chapter is present about Oxidative stress and cancer. This aspect must be examined more in depth.
(Response): Thanks. The chapter is supplemented with additional information in depth.
To avoid repeat of the same concepts results and discussion sections must be combined. In this way, the side effects of the therapies (the context) and the possible role of the plant extracts in that context can be discussed and understood. A further classification of the side effects of the therapies and the corresponding herbal formula or plant extract can improve the readability, otherwise it sounds simply like a list. I mean after discussing the cancer therapy (for example chemotherapy) a paragraph for the side effects (emesis, hepatotoxicity, skeletal muscle dysfunction, etc) and their corresponding “therapeutic support” should be added.
(Response): You are right. The manuscript has been revised according to your comments.
The tables are useful, but the formatting must be ameliorated.
(Response): Thank you for the comment. The table is revised.
A reference should be added: after lines 111-114; after lines 121-127.
(Response): Sorry for the confusion caused. The references were added.
Bibliography must be checked: journal abbreviations are often incorrect; the pages are often lacking. Check reference number: 2, 17-19, 24-25, 28-31, 33, 42-43, 45, 54, 64, 75, 89, 91, 96-105, 108-118, 123, 129-132, 138.
(Response): Thanks. We used MDPI journal endnote style however, there are some errors in references. All the references were check again and revised as Antioxidants guideline.
Extensive editing of English language and style is required.
(Response): Now the manuscript is edited by English native speaker.
Just as an example: after the period do not start with And (see lines 169, 180, 428); damage with DNA at line 99; at line 83 change use with uses and at line 81 add in It is usually used addition;
(Response): Revised.
Typos: Line 138 caner is cancer; lines 196 and 197 Cepharanthine hydrochloride
(Response): Revised.
Add a space at lines 173 creative protein; Withaniasomnifera at line 179;
(Response): Edited.
At line 240 substitute D.M. with Surname..
(Response): Modified.
At line 256 Ana Paula is the name, use the Surname
(Response): Thanks. Changed.
At line 332 was treated to the patients in incorrect: the patients were treated.
(Response): Revised.
At line 379 the following treatment.
(Response): Revised.
At line 577 remove were before randomly received
(Response): Revised.
At lines 578-579 “Cancer-related fatigue is one of the common cachexia. symptoms?
(Response): Revised.
Again we appreciate reviewers and editors for their kind and careful comments for improving the quality of our manuscript and also sincerely hope we address our responses well to the raised comments and our revised manuscript would be accepted for publication in your journal soon.
With kind regards,
Prof. Bonglee Kim, M.D, Ph.D.
Department of Pathology, College of Korean Medicine, Kyung Hee University
1 Hoegi-dong, Dongdaemun-ku, Seoul 130 -701, South Korea
E-mail: [email protected]
Tel; +82-2-961-9217, Fax; +82-2-961-9217
Reviewer 2 Report
This manuscript detailed introduction to exploring the role of as possible agents for sequela of cancer and cachexia of plant extracts. This review detail and highlights the seventy-four studies regarding plant extracts showing ability to manage the sequela and cachexia and antioxidant properties. This manuscript content is suitable for publication in Antioxidants.
Author Response
We appreciate editors and reviewers for critical comments to improve the quality of our manuscript (antioxidants-913920), titled “Plant extracts as possible agents for sequela of cancer therapies and cachexia”. We earnestly responded to the raised comments point by points.
This manuscript detailed introduction to exploring the role of as possible agents for sequela of cancer and cachexia of plant extracts. This review detail and highlights the seventy-four studies regarding plant extracts showing ability to manage the sequela and cachexia and antioxidant properties. This manuscript content is suitable for publication in Antioxidants.
(Response): Thank you very much for your comments. It means a lot to us. We hope this manuscript is helpful for the researchers and the cancer patients.
Again we appreciate reviewers and editors for their kind and careful comments for improving the quality of our manuscript and also sincerely hope we address our responses well to the raised comments and our revised manuscript would be accepted for publication in your journal soon.
With kind regards,
Prof. Bonglee Kim, M.D, Ph.D.
Department of Pathology, College of Korean Medicine, Kyung Hee University
1 Hoegi-dong, Dongdaemun-ku, Seoul 130 -701, South Korea
E-mail: [email protected]
Tel; +82-2-961-9217, Fax; +82-2-961-9217
Reviewer 3 Report
The paper is of interest and well written.
I think that the 2 main tables (3 and 5) need to be shortened and modified. I would summarize the "efficacy part" in order to make the tables more readable.
I would ask also to make an hypothesis regarding a possible trial to be developed for cancer patients, starting from this preclinical evidence. It can be discussed into the discussion part. Would it be a single or double arm trial? Would it make sense to compare the experimental arm with placebo? What would be the primary endpoint of the trial?
Author Response
We appreciate editors and reviewers for critical comments to improve the quality of our manuscript (antioxidants-913920), titled “Plant extracts as possible agents for sequela of cancer therapies and cachexia”. We earnestly responded to the raised comments point by points.
The paper is of interest and well written.
I think that the 2 main tables (3 and 5) need to be shortened and modified. I would summarize the "efficacy part" in order to make the tables more readable.
(Response): Thanks. All the tables are ameliorated and summarized for readers.
I would ask also to make an hypothesis regarding a possible trial to be developed for cancer patients, starting from this preclinical evidence. It can be discussed into the discussion part. Would it be a single or double arm trial? Would it make sense to compare the experimental arm with placebo? What would be the primary endpoint of the trial?
(Response): Thank you for the comments. The “hypothesis of a possible trial” part is added in discussion.
Again we appreciate reviewers and editors for their kind and careful comments for improving the quality of our manuscript and also sincerely hope we address our responses well to the raised comments and our revised manuscript would be accepted for publication in your journal soon.
With kind regards,
Prof. Bonglee Kim, M.D, Ph.D.
Department of Pathology, College of Korean Medicine, Kyung Hee University
1 Hoegi-dong, Dongdaemun-ku, Seoul 130 -701, South Korea
E-mail: [email protected]
Tel; +82-2-961-9217, Fax; +82-2-961-9217v
Round 2
Reviewer 1 Report
The current version is fine. Please check table 3 formatting.